# External signals regulate continuous transcriptional states in hematopoietic stem cells

Eva M Fast[1], Audrey Sporrij[1], Margot Manning[1], Edroaldo Lummertz Rocha[2], Song Yang[3], Yi Zhou[3], Jimin Guo[4], Ninib Baryawno[5], Nikolaos Barkas[6], David Scadden[7], Fernando Camargo[8], Leonard I Zon[9]*

[1]Department of Stem Cell and Regenerative Biology, Harvard University, Cambridge, United States; [2]Laboratório de Imunobiologia, Departmento de Microbiologia, Imunologia e Parasitologia, Universidade Federal de Santa Catarina, Florianópolis, Brazil; [3]Stem Cell Program and Division of Hematology/Oncology, Howard Hughes Medical Institute, Boston's Children's Hospital and Dana Farber Cancer Institute, Harvard Medical School, Boston, United States; [4]Medical Devices Research Centre, National Research Council Canada, Boucherville, Canada; [5]Childhood Cancer Research Unit, Department of Children's and Women's Health, Karolinska Institutet, Stockholm, Sweden; [6]Broad Institute of Harvard and MIT, Cambridge, United States; [7]Harvard University, Cambridge, United States; [8]Children's Hospital Harvard Med Sch, Cambridge, United States; [9]Stem Cell Program and Hematology/Oncology, Boston Children's Hospital, Boston, United States

*For correspondence:
zon@enders.tch.harvard.edu

**Abstract** Hematopoietic stem cells (HSCs) must ensure adequate blood cell production following distinct external stressors. A comprehensive understanding of in vivo heterogeneity and specificity of HSC responses to external stimuli is currently lacking. We performed single-cell RNA sequencing (scRNA-Seq) on functionally validated mouse HSCs and LSK (Lin-, c-Kit+, Sca1+) progenitors after in vivo pharmacological perturbation of niche signals interferon, granulocyte colony-stimulating factor (G-CSF), and prostaglandin. We identified six HSC states that are characterized by enrichment but not exclusive expression of marker genes. External signals induced rapid transitions between HSC states but transcriptional response varied both between external stimulants and within the HSC population for a given perturbation. In contrast to LSK progenitors, HSCs were characterized by a greater link between molecular signatures at baseline and in response to external stressors. Chromatin analysis of unperturbed HSCs and LSKs by scATAC-Seq suggested some HSC-specific, cell intrinsic predispositions to niche signals. We compiled a comprehensive resource of HSC- and LSK progenitor-specific chromatin and transcriptional features that represent determinants of signal receptiveness and regenerative potential during stress hematopoiesis.

## Introduction

Stem cell therapy holds promises for numerous indications, including blood diseases, autoimmune diseases, neurodegeneration, and cancer (*Blau and Daley, 2019*). Despite being used in the clinic for over 30 years, hematopoietic stem cell (HSC) transplants remain a highly risky procedure. To better understand HSC regeneration, recent efforts have used single-cell RNA sequencing (scRNA-Seq) to discover novel markers to further enrich for functional HSCs (*Chen et al., 2016*; *Cabezas-Wallscheid et al., 2017*; *Wilson et al., 2015*; *Rodriguez-Fraticelli et al., 2020*). Yet, no consensus exists on the optimal marker combination to obtain the most purified HSCs in part because extensive functional

**eLife digest** Most organs in the human body are maintained by a type of immature cells known as adult stem cells, which ensure a constant supply of new, mature cells. Adult stem cells monitor their environment through external signalling molecules and replace damaged cells as needed.

Stem cell therapy takes advantage of the regenerative ability of immature stem cells and can be helpful for conditions such as blood diseases, autoimmune diseases, neurodegeneration and cancer. For example, hematopoietic stem-cell transplantation is a treatment for some types of cancer and blood disorders, in which stem cells are harvested from the blood or bone marrow and reintroduced into the body, where they can develop into all types of blood cells, including white blood cells, red blood cells and platelets.

Hematopoietic stem-cell transplants have been in use for over 30 years, but they remain a highly risky procedure. One of the challenges is that outcomes can vary between patients and many of the factors that can influence the 'regenerative' potential of hematopoietic stem cells, such as external signalling molecules, are not well understood.

To fill this gap, Fast et al. analysed which genes are turned on and off in hematopoietic stem cells in response to several external signalling molecules. To do so, three signalling pathways in mice were altered by injecting them with different chemicals. After two hours, the hematopoietic stem cells were purified and the gene expression for each cell was analysed.

This revealed that the types of genes and the strength at which they were affected by each chemical was unique. Moreover, hematopoietic stem cells responded rapidly to external signals, with substantial differences in gene expression between individual groups of cells. Contrary to more specialised cells, the external signalling genes in some hematopoietic stem cells were already activated without being injected with external signalling molecules. This suggest that low levels of external signalling molecules released from their microenvironment may prepare stem cells to better respond to future stress or injuries.

These results help to better understand stem cells and to evaluate how the signalling state of hematopoietic stem cells affects regeneration, and ultimately improve hematopoietic stem cell transplantation for patients.

heterogeneity within HSCs makes experimental evaluation challenging (*Haas et al., 2018*). Both intrinsic and extrinsic factors have been implicated in regulating HSC function (*Zon, 2008*; *Morrison et al., 1996*). The stem cell niche forms an important extrinsic regulator of HSCs as it anchors stem cells and maintains the balance between self-renewal and differentiation (*Morrison and Spradling, 2008*; *Morrison and Scadden, 2014*). Release of soluble signals from the niche such as interferons, prostaglandins, and growth factors, including stem cell factor (SCF) and G-CSF, has been shown to influence HSC function during homeostasis and upon injury (*Pinho and Frenette, 2019*; *Pietras et al., 2016*; *Zhao et al., 2014*; *Morales-Mantilla and King, 2018*). While known to be affected by a wide variety of extracellular signals, little is known about the heterogeneity and specificity of HSC responses to these external stimuli, nor is it understood how differential responses relate to functional diversity of HSCs. HSCs are also regulated cell intrinsically (*Zon, 2008*; *Morrison et al., 1996*). Chromatin state is a crucial determinant of cell identity and behavior (*Klemm et al., 2019*). Hematopoietic differentiation is a prime example of how cell fate changes associate with massive remodeling of the epigenetic landscape (*Avgustinova and Benitah, 2016*). Despite the current knowledge on regulators of HSC fate, few studies have assessed chromatin states in purified, in vivo-derived HSC populations (*Yu et al., 2017*; *Lara-Astiaso et al., 2014*) due to technical limitations such as cell numbers. Recent advancements in single-cell chromatin accessibility sequencing (scATAC-Seq) provides a methodological framework for studying the diversity and uniqueness of HSC chromatin features at homeostasis and upon external stimulation (*Buenrostro et al., 2018*; *Lareau et al., 2019*).

Here, we performed comprehensive scRNA-Seq and scATAC-Seq profiling on functionally validated mouse HSCs and examined in vivo transcriptional responses to pharmacological stimulation, mimicking signals from the stem cell niche. To encompass a wide variety of different transcriptional responses, we evaluated three different signaling pathways: an inflammatory pathway through stimulation or inhibition of prostaglandins by 16,16-dimethyl prostaglandin $E_2$ (dmPGE$_2$) and indomethacin,

a host-defense immune signaling pathway mediated by activating of TLR and interferon signaling with poly(I:C), and a cellular mobilization pathway stimulated by the growth factor G-CSF. We found that unperturbed HSCs exist in fluent transcriptional states with different levels of marker gene enrichment. External stimulants can alter the cell distribution between HSC states to varying degrees depending on the stimulant as well as induce specific changes within cell states. Comparison of HSCs to multipotent LSK (Lin-, c-Kit+, Sca1+) progenitors allowed us to determine the specificity of transcriptional responses in HSCs. Finally, analysis of native HSC chromatin states revealed cell intrinsic heterogeneity that may prime HSC subpopulations for particular transcriptional responses following exposure to certain signals. The data is provided as a resource to the broader research community via an easily accessible web interactive application (https://mouse-hsc.cells.ucsc.edu). This work provides a comprehensive description of the in vivo single-cell transcriptomic and epigenetic landscape of HSCs and multipotent LSK progenitors in response to common external stressors.

## Results

### In vivo stimulation of functionally validated HSCs and multipotent progenitors for transcriptomic and epigenetic profiling

To investigate transcriptional responses to external signals, we profiled HSCs and multipotent progenitors (MPPs) after four distinct in vivo pharmacological perturbations with doses matching previous studies (*Figure 1A*, see Materials and methods). Male and female mice were treated with one of three activators dmPGE$_2$, poly(I:C), or G-CSF for 2 hr or administered the Cox1/2 inhibitor indomethacin ('Indo') for 1 week to deplete endogenous prostaglandins (see Materials and methods). We chose a 2 hr treatment window for the extrinsic activators as we aimed to assess the immediate, direct effects of the external stimulants on HSCs and MPPs. After the respective drug treatments, HSC and MPP populations comprising the entire LSK compartment were isolated via fluorescence-activated cell sorting (FACS) (*Figure 1—figure supplement 1A*). Through a limiting dilution transplantation assay (LDTA) and extreme limiting dilution assay (ELDA) analysis (*Hu and Smyth, 2009*), we determined HSC purity to be 1 in 8 (*Figure 1—figure supplement 1B-D*). The LDTA confirmed that our isolation and purification procedure allowed for the profiling of functional, highly purified HSCs. Phenotypic marker composition within LSK cells remained largely consistent between different stimulations (*Figure 1—figure supplement 1E*). An exception was the reduction of cells within the HSC compartment following dmPGE$_2$ treatment, decreasing from 1.9% in control to 0.85% of LSK cells (p-value = $6.4*10^{-4}$, by differential proportion analysis [DPA]; *Farbehi et al., 2019*). To account for a potential phenotypic shift in HSC surface marker expression due to CD34 externalization, which would move functional HSCs to the MPP1 population, we compared the contribution of the later by scRNA-Seq-defined 'stem cell state' in HSCs and MPP1s. We found no increase in the 'stem cell' population in dmPGE$_2$-treated MPP1s, compared to the control (*Figure 1—figure supplement 2H*). After cell sorting, we subjected a total of 46,344 cells to scRNA-Seq using the 10× Genomics platform (see Materials and methods). We obtained an average of 37,121 (SD = 14,308) reads per cell and 2994 (SD = 480) genes per cell (*Supplementary file 1*), indicative of a rich dataset that contained functionally validated HSCs.

### Continuous transcriptional states in HSCs at baseline

To determine how external stimulants affect specifically HSCs in vivo, we first analyzed a combination of highly purified control and treated HSCs but not MPPs cells (*Figure 1A*). We applied a standard scRNA-Seq pipeline to filter and normalize UMI reads (see Materials and methods). Separate analysis of male and female HSCs revealed minimal sexual dimorphism during both steady state and following perturbation with external stimulants (*Figure 1—figure supplement 3*, *Supplementary files 2 and 3*). We therefore regressed out any sex-specific effects and controlled for other batch-specific confounders in further downstream analyses (see Materials and methods). In the aggregated dataset, we detected a total of six HSC clusters (*Figure 1B*). To ensure optimal choice of clustering hyperparameters, we used a data-driven approach (Silhouette coefficient and Davies–Bouldin index) that was validated by comparison of two independent biological scRNA-Seq replicates of control HSCs sorted from different mouse strains (see Materials and methods, *Figure 1—figure supplement 2A-D*, *Supplementary file 4*). The absence of clear separation into highly distinct clusters in uniform manifold approximation and

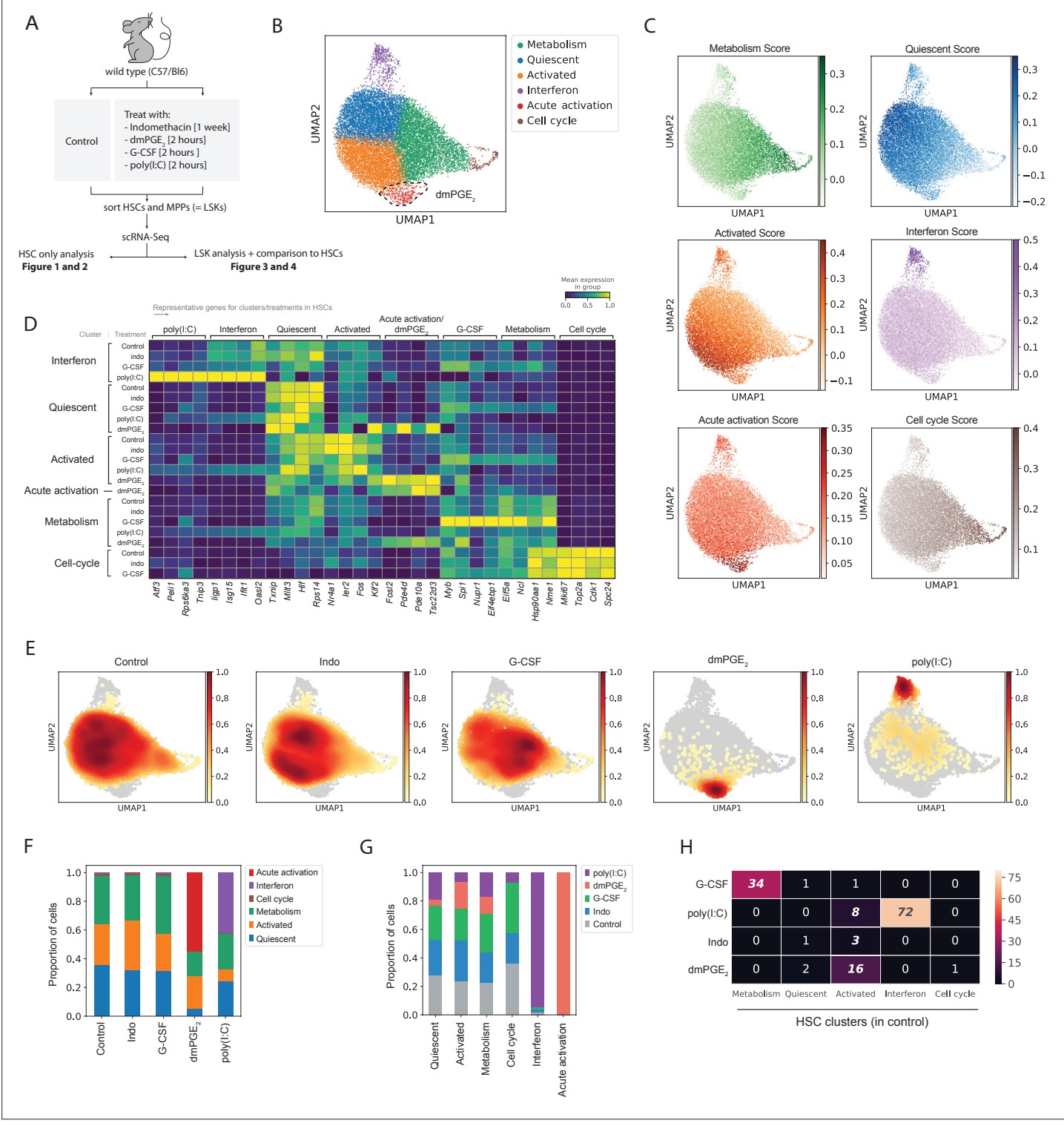

**Figure 1.** Hematopoietic stem cells (HSCs) are transcriptionally heterogeneous and niche perturbations rapidly shift cells into different states. (**A**) Schematic of stimulant treatment before HSC and multipotent progenitor (MPP) isolation, see also *Figure 1—figure supplement 1*. (**B**) Uniform manifold approximation and projection (UMAP) plot of HSC clusters (n = 15,355 cells), with 16,16-dimethyl prostaglandin E$_2$ (dmPGE$_2$)-induced cluster (red) traced with a dashed line, see also *Figure 1—figure supplement 2A-G*. (**C**) UMAP plot with transcriptional scores for each cluster. (**D**) Heatmap of selected enriched genes for each HSC cluster and treatment (columns, scaled expression) averaged gene expression for all cells within a cluster and treatment (rows, only clusters shown with >20 cells), see also *Figure 1—figure supplement 2I* and *Figure 1—figure supplement 4*. (**E**) UMAP density graphs of HSC distribution for each external stimulant. (**F**) Proportion of HSCs within clusters for each perturbation. (**G**) Proportion of HSCs of each

*Figure 1 continued on next page*

*Figure 1 continued*

perturbation within a cluster normalized for total cell number per treatment. (**H**) Heatmap with number of common genes between the 100 top induced genes per HSC treatment (rows) and HSC clusters (columns), false discovery rate (FDR)-corrected hypergeometric p-values < 0.01 are italicized, exact p-values in *Figure 1—source data 1*. For separate analysis of male and female HSCs, see *Figure 1—figure supplement 3*.

The online version of this article includes the following figure supplement(s) for figure 1:

**Source data 1.** Excel spreadsheet containing quantitative data for *Figure 1*.

**Figure supplement 1.** Functional characterization of hematopoietic stem cells (HSCs) confirms high regenerative capacity.

**Figure supplement 2.** Evaluation of single-cell RNA sequencing (scRNA-Seq) clustering with independent replicates, candidate genes, and transcriptional scores.

**Figure supplement 3.** Minimal sexual dimorphism in hematopoietic stem cells (HSCs) and Lin-, c-Kit+, Sca1+ (LSKs) in steady state and upon stimulation.

**Figure supplement 4.** Heatmaps of differentially expressed genes in hematopoietic stem cells (HSCs) enables identification of genes and single-cell clusters with similar expression patterns.

projection (UMAP) space (*Figure 1B*), together with fact that most marker genes were not exclusively expressed but rather enriched in a given cluster (*Figure 1—figure supplement 2E*), suggests that the HSC clusters represent transcriptional states with continuous transitions as opposed to discrete subtypes of HSCs. We calculated a transcriptional score by combining the top enriched genes for each cluster (*Figure 1C*, see Materials and methods) to further illustrate the observation of gradual changes in transcriptional state within the HSC population. While transcriptional scores were most enriched in their respective clusters, expression dropped before and extended beyond cluster borders (*Figure 1B and C*). Reactome and gene ontology (GO) term pathway enrichment analysis, comparison to previous studies of functionally characterized HSCs (Materials and methods, *Supplementary files 5 and 6*) and manual curation of enriched genes (*Figure 1D*, *Figure 1—figure supplement 2E*, *Supplementary file 4*) allowed to assign labels to each HSC cluster or state. Three HSC clusters made up 98% of control HSCs (*Figure 1F*) while the remaining 2% split into a 'cell cycle' cluster marked by genes such as *Ki67* and an 'Interferon' cluster characterized by the expression of interferon-response genes *Iigp1*, *Isg15*, *Ifit1*, *and Oasl2* (each 1%, *Figure 1D and F*). A prominent HSC subpopulation was defined by various immediate early genes (IEGs) including *Nr4a1, Ier2,* and *Fos* (*Figure 1D* and *Figure 1—figure supplement 2E*) and we therefore named this cluster 'Activated'. We eliminated the possibility that the 'Activated' cluster arose due to an unspecific artifact of the cell isolation procedure since LSKs did not have an 'Activated' cluster and the proportion of *Nr4a1* expressing cells was much smaller (Figure 3B and Figure 3—figure supplement 1B). HSCs have been tightly associated with decreased cell cycle activity (*Foudi et al., 2009*; *Wilson et al., 2008*; *Qiu et al., 2014*). The cluster adjacent to the 'Activated' state was termed 'quiescent' because cells showed enrichment in expression of marker genes that have previously been linked to the most potent and quiescent HSCs (*Figure 1D*, *Figure 1—figure supplement 2F*, *Supplementary file 6*; *Cabezas-Wallscheid et al., 2017*; *Chen et al., 2016*; *Wilson et al., 2015*; *Acar et al., 2015*; *Gazit et al., 2014*; *Balazs et al., 2006*; *Komorowska et al., 2017*; *Schneider et al., 2016*; *Jeong et al., 2009*). Furthermore, 'quiescent' HSCs did not express IEGs and expressed low levels of the 'cell cycle' score (*Figure 1B and C*). The 'metabolism' cluster comprised the most metabolically active HSCs as evidenced by enrichment of transcripts involved in translation initiation (*Eif5a, Eif4a1*), nucleotide metabolism (*Nme1, Dctpp1*), ribosome assembly (*Ncl, Nop56, Nop10, Npm1*) and protein chaperones (*Hsp90, Hsp60*) (*Figure 1B and D*, *Supplementary file 4*). In conclusion, baseline HSCs were defined by three main transcriptional states, 'Quiescent', 'Activated', and 'Metabolism' (*Figure 1F*) with few HSCs residing in the 'Interferon' or 'Cell cycle' state. Transcriptional scores visualized that these HSC states were not exclusive and that HSC transcriptional state could be rather described by a combination of continuous gradients of marker genes. Therefore, subsequent analyses via discrete clusters provided an analytical tool to compare changes in transcriptional state as opposed to an exclusive assignment of cell identities.

## External signals changed HSC distribution between clusters and transcriptional activity within clusters

To determine how external stimulants affect transcriptional identity of HSCs, we evaluated changes in cell distribution between clusters (*Figure 1E and F*) as well as differentially expressed genes (DEGs)

within each cluster using 'model-based analysis of single-cell transcriptomics' or MAST (see Materials and methods; *Finak et al., 2015*, *Supplementary file 7*). We further examined the relationship of genes that define each HSC cluster and genes perturbed by each external stimulant (*Figure 1D and H*). A unified heatmap shows all HSC clusters for every perturbation (rows) and the averaged gene expression within these clusters for four cluster- or treatment-representative genes (columns, up-only, *Figure 1D*, *Supplementary files 4 and 8*, full heatmap in *Figure 1—figure supplement 2I*). To further identify distinct patterns of gene regulation in HSC clusters and visualize both up- and downregulated genes, we generated separate heatmaps for each individual perturbation (*Figure 1—figure supplement 4*, *Supplementary file 8*). DmPGE₂ and poly(I:C) stimulated genes showed enrichment for previously described signatures with the same stimulants (*Supplementary files 5 and 6*). G-CSF induced selected genes such as *Myb* and *Spi1* (*Figure 1D*) and downregulated niche adhesion receptors *ckit* and *Cd9* (*Figure 1—figure supplement 4B*, purple arrows) consistent with the growth factor's role in myeloid differentiation (*Metcalf and Nicola, 1983*) and mobilization (*Leung et al., 2011*; *Bendall and Bradstock, 2014*), respectively. However, our G-CSF-induced gene set did not show any significant enrichment (*Supplementary files 5 and 6*) with various previously reported G-CSF signatures (*Schuettpelz et al., 2014*; *Pedersen et al., 2016*; *Giladi et al., 2018*; *Mervosh et al., 2018*) likely due to different timing of G-CSF treatment. Indomethacin only led to subtle changes in gene expression (*Figure 1—figure supplement 4D*, *Supplementary file 8*) and cell distribution between HSC clusters remained unaffected (*Figure 1F*). Both dmPGE₂ and poly(I:C) caused a significant change in HSC cluster distribution which indicated a loss of the original transcriptional identity of some HSCs (*Figure 1D–F*). In vivo treatment with dmPGE₂ gave rise to a novel cluster that contained 55% of dmPGE₂-treated HSCs (*Figure 1F*) and which was itself only composed of dmPGE₂-treated cells (*Figure 1G*). We called this cluster 'Acute activation' (*Figure 1B*) since marker genes included known cAMP-response genes such as *Fosl2* (*Figure 1D* and *Figure 1—figure supplement 2E*) and the phosphodiesterases *Pde10a*, *Pde4b*, and *Pde4d* (*Figure 1D*, *Supplementary file 4*). The 'Acute activation' cluster displayed the highest transcriptional score of marker genes from the 'Activated' cluster (besides the 'Activated' cluster itself) including genes such as *Klf2* which confirmed the close relationship between these two clusters (*Figure 1D and H* and *Figure 1—figure supplement 2G*, p-value [Tukey's honest significant differences, HSD] = 0.001). dmPGE₂-treated cells in other clusters also showed strong expression of target genes such as *Tsc22d3*, but in contrast to the 'Acute activation' cluster the expression of cluster identity genes (e.g. *Txnip*, *Mllt3*) was maintained in the dmPGE₂-treated 'quiescent' cluster (*Figure 1D*). Poly(I:C) treatment increased the proportion of HSCs in the 'interferon' cluster from 1% to 42% (*Figure 1F* and p-value [DPA] $<10^{-5}$). The top 100 poly(I:C)-stimulated genes exhibited a 72% overlap with the top 100 marker genes of the 'interferon' cluster (*Figure 1H* and p-value (hypergeometric test, false discovery rate [FDR]-corrected) = $10^{-144}$, *Supplementary file 9*) suggesting that poly(I:C) treatment reinforces a transcriptional program that already exists endogenously in a small proportion of HSCs (*Figure 1*, *Figure 1—figure supplement 2A-D*). In contrast to dmPGE₂, the transcriptional response to poly(I:C) was strongest in the 'interferon' cluster since target genes, for example, *Oasl2* or *Peli1*, were less induced in the other poly(I:C)-treated clusters (*Figure 1D*). Treatment with G-CSF led only to minimal shifts in HSC distribution (*Figure 1E*) and proportions between HSC clusters, respectively (*Figure 1F* and p-value [DPA] >0.05 for all clusters). The transcriptional response for most G-CSF target genes such as *Myb*, *Eif4ebp1*, *or Ncl* was strongest within the 'metabolism' cluster (*Figure 1D*) with a 34% overlap (p-value [hypergeometric test, FDR-corrected] = $8.2*10^{-49}$) between 'metabolism' marker genes and G-CSF-induced genes (*Figure 1H*, *Supplementary file 9*). In summary a 2 hr in vivo pulse with poly(I:C) or dmPGE₂ significantly altered distributions of HSCs between pre-existing transcriptional states and, in the case of dmPGE₂, allowed for a novel transcriptional state to surface. The fact that certain clusters (e.g. 'metabolism' and 'interferon') responded more strongly to external stimuli combined with the observation that HSCs kept their baseline cluster identity to varying degrees strongly suggests that transcriptional heterogeneity does not only exist at baseline but also during HSCs' response to extrinsic signals.

## Endogenous cell states distinguished TLR- and IFN-specific responses of poly(I:C) treatment

To better understand how poly(I:C) induced interferon signaling, we evaluated different components of the TLR and interferon pathways in our single-cell clusters. Binding of poly(I:C) to Toll-like receptor

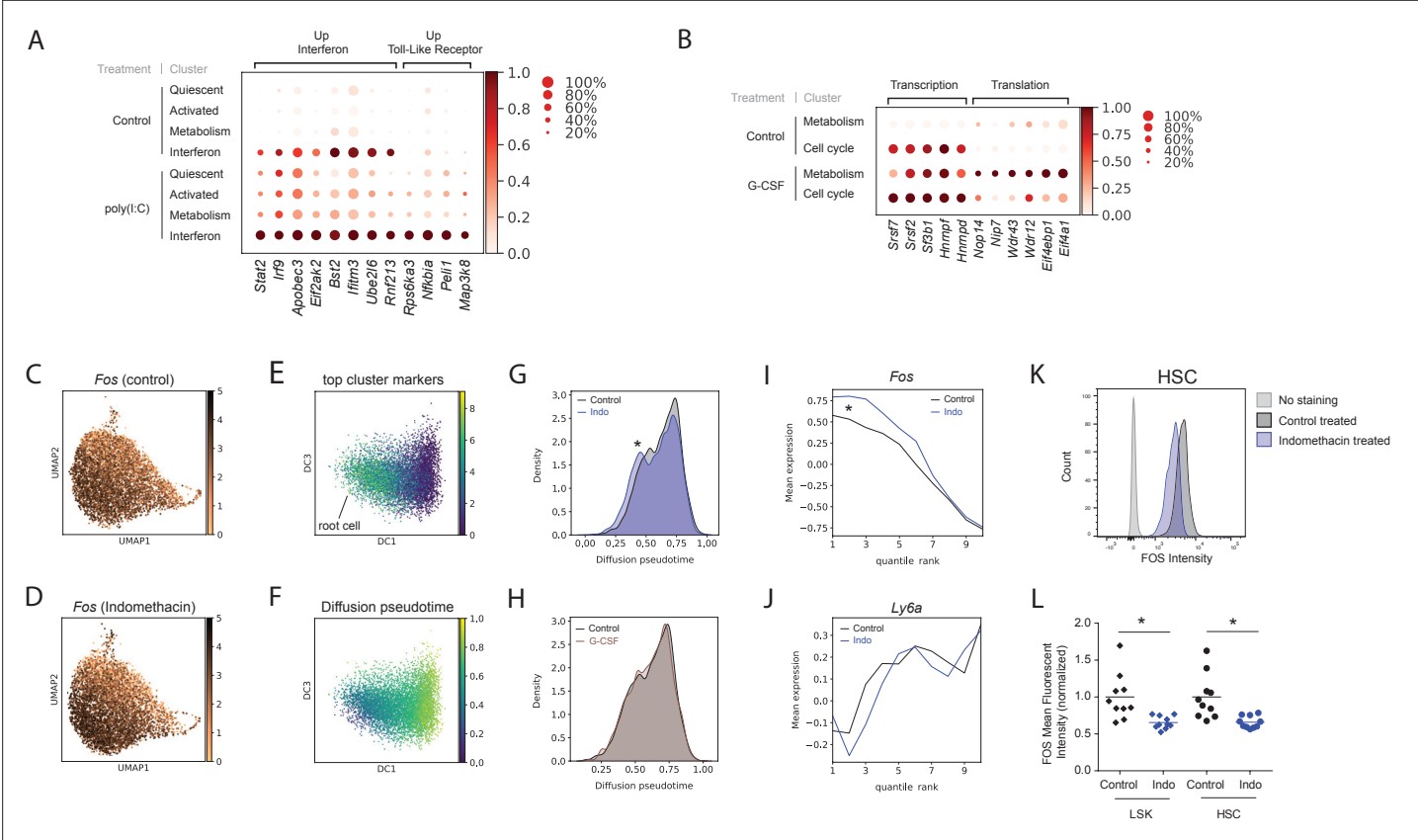

**Figure 2.** Poly(I:C), granulocyte colony-stimulating factor (G-CSF), and indomethacin induce cluster-specific transcriptional changes in hematopoietic stem cells (HSCs). (**A**) Dot plot of representative genes from poly(I:C) treated and control HSC clusters (scaled expression across columns). (**B**) Dot plot of representative genes from the G-CSF-treated and control HSC clusters (scaled expression across columns). (**C–J**) Diffusion pseudotime analysis. Uniform manifold approximation and projection (UMAP) plot of *Fos* expression in control (**C**) and upon indomethacin (**D**) treatment, see also *Figure 2—figure supplement 1A-B*. Diffusion map embedding with combined expression of top 'Activated' genes to select root cell (**E**) and cells colored by pseudotime (**F**). Kernel density of pseudotime distribution comparing indomethacin and control (**G**, asterisk: p-value [Mann–Whitney U-test] = 5.8*10⁻¹²) and G-CSF and control (**H**). Average expression of *Fos* (**I**) and *Ly6a* (**J**) across cells ranked by pseudotime (cells split into 10 bins to decrease noise), change in transcript levels indicated by asterisk in I, see also *Figure 2—figure supplement 1C-D*. (**K**) Histogram of FOS levels via intracellular fluorescence-activated cell sorting (FACS) of HSCs, 'no stain' is FACS-negative control, 'control' is FOS in untreated mice. (**L**) Normalized mean fluorescent intensity (MFI) for FOS in control and indomethacin-treated HSCs (p-value = 6.2 * 10⁻³, Welch-corrected t-test, asterisk) and LSK cells (p-value = 6.6 * 10⁻³, Welch-corrected t-test, asterisk) across two independent biological replicate experiments, n(mice) = 20.

The online version of this article includes the following figure supplement(s) for figure 2:

**Source data 1.** Excel spreadsheet containing quantitative data for *Figure 2*.

**Figure supplement 1.** Indomethacin affects transcriptional state of immediate early genes (IEGs).

3 (TLR3) (*Alexopoulou et al., 2001*) induces expression of Type I interferons (IFNα and IFNβ), which in turn signal via IFNα/β receptor 1 (*Ifnar1*) and 2 (*Ifnar2*) heterodimers, all of which were expressed in HSCs (Figure 4E). We identified two expression patterns in poly(I:C)-treated HSCs that were consistent with TLR and interferon receptor signaling. The first expression pattern 'up interferon' was driven by induction of poly(I:C) responsive genes across all cell states. In addition, these genes were already specifically enriched in the 'interferon' cluster in the absence of poly(I:C) stimulation (*Figure 2A*). Genes within this group are either directly downstream of Type I interferon receptors, such as *Stat2* and *Irf9*, or act as effector proteins involved in viral interferon response such as *Apobec3* and *Eif2ak2* (*Figure 2A*, *Figure 1—figure supplement 4A*). The high expression of several interferon-induced viral-response genes (e.g. *Bst2*, *Ifitm3*, *Ube2l6*, and *Rnf213*) in the control 'interferon' cluster might point to a state of general surveillance for viral infection at baseline (*Figure 2A*, *Figure 1—figure supplement 4A*). The second expression pattern 'up Toll-like receptor' constituted poly(I:C)-induced genes that were predominantly found in the 'interferon' cluster with low expression at baseline in the

control 'interferon' cluster (*Figure 2A*, *Figure 1—figure supplement 4A*). Genes within this signature included *Nfkbia*, *Peli1*, *Map3k8*, and *Rps6ka3* all of which are part of TNFα and Toll-like signaling pathways. This expression profile might therefore represent a more direct response to poly(I:C) interaction with Tlr3. Comparison of differential expression patterns across cell states allowed us to distinguish between poly(I:C)-mediated TLR- and interferon-based signaling.

## G-CSF triggered changes within the 'metabolism' cluster without changing cell distributions between clusters

Even though G-CSF did not change cell distribution between clusters (*Figure 1F*), it induced DEGs, most within the HSC 'metabolism' cluster (*Figure 1D*, *Figure 1—figure supplement 4B*). Hierarchical clustering suggested that G-CSF treatment drove the expression profile of the HSC 'metabolism' cluster closer toward the 'cell cycle' state (*Figure 1—figure supplement 4B*). This shift was facilitated by induction of genes related to transcription, such as RNA binding proteins (*Hnrnpd*, *Hnrnpf*, *Hnrnpa2b1*), as well as splicing factors (*Srsf7*, *Sf3b1*, *Srsf2*) ('transcription', *Figure 2B*). G-CSF also increased expression of transcripts involved in translation (ribosome biogenesis: *Nop14*, *Nip7*, *Wdr43*, *Wdr12* and translation initiation: *Eif4a1*, *Eif4ebp1*) that were not expressed in the 'cell cycle' state at baseline ('translation', *Figure 2B*). This may indicate a G-CSF-induced fate commitment toward differentiation. Overall, a 2 hr pulse of G-CSF pushed HSCs toward a more metabolically active state. Our scRNA-Seq data are consistent with the original description of G-CSF as a growth factor that regulates myeloid differentiation and indicates an early transcriptional response leading to HSC mobilization.

## Endogenous prostaglandins, perturbed by indomethacin, regulated IEGs within the 'Activated' cell state

To investigate external signaling in a more physiological setting, we orally treated mice for 1 week with indomethacin to deplete endogenous prostaglandins. Differential expression analysis identified only 21 genes (1.2-fold change cutoff, Figure 4C) affected by indomethacin. Ten out of twelve upregulated genes can be classified as IEGs (e.g. *Fos*, *Fosb*, *Jun*, *Klf4*, *or Klf6*) (*Figure 1—figure supplement 4D*, *Supplementary file 8*). While cell proportions did not change between the HSC clusters (*Figure 1F*), distribution of cells shifted slightly toward the periphery of the UMAP plot (*Figure 1E*) which was mirrored by increased expression of individual 'Activated' cluster marker genes such as *Fos* and other IEGs (*Figure 2C–D* and *Figure 2—figure supplement 1A-B*). To further investigate the influence of endogenous prostaglandin depletion on cell state while taking the entire transcriptional landscape into account, we computed diffusion pseudotime (DPT) (*Haghverdi et al., 2016*) between the 'Activated' and 'Quiescent' cluster in HSCs. The cell with the combined highest expression of the three top cluster markers for the 'Activated' state (*Figure 2E*, see Materials and methods) was set as the 'root cell' and DPT was calculated originating from that root cell (*Figure 2F*). Indomethacin-treated cells displayed a significant shift in overall pseudotime kernel density distribution, which is indicative of overall lower pseudotime (*Figure 2G*, shift indicated by asterisk, p-value = $5.8*10^{-12}$ by Mann–Whitney U-test). No shift was observed when comparing the control to G-CSF-treated HSCs (*Figure 2H* and p-value = 0.18). Ranking cells for each treatment condition according to pseudotime and averaging gene expression in 10 equally sized bins (quantile ranks 1–10) further illustrated the change in expression of *Fos* and other IEG genes following indomethacin, especially at lower pseudotimes (*Figure 2I* and *Figure 2—figure supplement 1C*; indicated by asterisks). Genes that were not part of the 'Activated' gene signature, such as *Ly6a*, did not follow the same pattern (*Figure 2J*), nor was a similar trend observed in response to G-CSF treatment (*Figure 2—figure supplement 1D*). The pseudotime analysis of the scRNA-Seq data indicated a specific shift in IEG transcriptional state upon depletion of endogenous prostaglandins. To further confirm the effect of endogenous prostaglandins on IEGs in an orthogonal assay, we measured single-cell protein levels of FOS by intracellular flow cytometry. Across two independent experiments, a 7-day in vivo indomethacin treatment led on average to a 34% (SD = 8.2%) reduction in FOS mean fluorescent intensity (MFI) in HSCs (p-value = $6.2 * 10^{-3}$, t-test with Welch's correction) and a mean 35% (SD = 8.6%) decrease in LSKs (p = $6.6 * 10^{-3}$, *Figure 2K–L*). Overall, endogenous prostaglandin levels impacted both the transcriptional state and protein levels of FOS and potentially other IEGs.

## Increased differentiation and cell cycle signatures within transcriptional states of LSKs compared to HSCs

To evaluate specificity of transcriptional heterogeneity observed within HSCs and their response to external signals, we analyzed the transcriptome of the entire LSK compartment, which encompasses mostly MPPs and a small proportion (~2%) of HSCs (*Figure 1—figure supplement 1A* and E). Transcriptional responses and LSK cell states in phenotypically defined MPPs (*Cabezas-Wallscheid et al., 2014*; *Pietras et al., 2015*) (MPP0, MPP1, MPP2, MPP3/4, *Figure 1—figure supplement 1A*) were profiled using a hashtag oligonucleotide (HTO) labeling strategy that is part of the cellular indexing of transcriptomes and epitopes by sequencing (CITE-Seq) methodology (*Figure 3—figure supplement 1A, C, D* and Materials and methods *Stoeckius et al., 2018*). Cell hashing enables tracking of cell surface phenotypes in scRNA-Seq data through barcoding of cells with antibody conjugated DNA-oligos (HTO barcoding). ScRNA-Seq gene expression of marker genes such as *Cd34*, *Cd48*, and *Cd150 (Slamf1)* matched the surface phenotypes used for sorting of HTO-barcoded MPPs, confirming that our workflow was successful (*Figure 3—figure supplement 1B, E*). We analyzed transcriptomic data from LSK cells as an aggregated set consisting of all four perturbations and control, analogous to the approach used for HSCs above. We discovered a total of eight LSK clusters, which similar to HSCs displayed gene expression enrichment as opposed to exclusive expression of marker genes (*Figure 3B*, *Figure 3—figure supplement 1B*). These LSK clusters were labeled by analysis of enriched genes and pathways (*Figure 3E*, *Supplementary files 4-6*), their composition of phenotypically defined cell populations tracked by HTO barcoding (*Figure 3—figure supplement 1C* and G) and by comparing the top 100 enriched genes of LSK clusters to the earlier defined HSC clusters (*Figure 3A*, *Supplementary file 9*). Because the latter analysis only indicated similarity rather than full equivalence of HSC and LSK clusters, and to avoid ambiguity when evaluating HSCs and LSKs, all LSK clusters were denoted with the prefix 'LSK-'. LSK clusters most similar to the 'quiescent' HSC state by top enriched genes were named 'LSK-primitive' and 'LSK-primed', respectively (*Figure 3A*). These two clusters further expressed the highest level of the HSC 'quiescence' score (*Figure 3—figure supplement 1H*, p-value(Tukey's HSD) = 0.001). The 'LSK-primitive' cluster encompassed the majority of phenotypic HSCs and was significantly depleted of MPP3/4s compared to all other clusters (*Figure 3—figure supplement 1F-G*, DPA p-values < 0.02). LSK cells in the 'LSK-primed' cluster represented a more committed state given their expression of *Cd34* and *Flt3*. Enrichment of *Cd37* and *Sox4* suggested priming toward a lymphoid fate (*Figure 3E*; *Sun et al., 2013*; *Zou et al., 2018*). In contrast to HSCs, a higher proportion of LSKs were in a metabolically active or cycling state (43% LSKs [*Figure 3C*] vs. 35% HSCs [*Figure 1F*], p-value (chi-squared test) = $1.7*10^{-5}$). In addition, the 'LSK-metabolism' cluster itself exhibited a stronger cell cycle signature compared to the HSC 'metabolism' cluster (*Figure 3A* and increased expression of *Ki67* and *Top2a Figure 3E* vs. *Figure 1D*). A small proportion of LSKs (<1%, *Figure 3B–C*), comprising the 'LSK-myeloid' cluster, were defined by expression of genes such as *Mpo*, *Ctsg*, *Fcer1g*, and *Cebpα* (*Figure 3E*). Consistent with previous reports (*Pietras et al., 2015*), our data indicated that the 'LSK-myeloid' cluster was composed of MPP2s and MPP3/4 cells but no HSCs, MPP0s, or MPP1s (*Figure 3—figure supplement 1G*). In summary, control-treated LSKs were distributed among four main clusters, those being 'LSK-primed', 'LSK-primitive', 'LSK-metabolism', and 'LSK-cell cycle', that together encompassed >99% of control LSK cells (*Figure 3C*). Comparison to HSC clusters and HTO-barcoded MPPs allowed to define identities of LSK clusters. Consistent with previous functional studies, we found enrichment of phenotypically defined MPPs in corresponding transcriptional clusters (e.g. MPP2 and -3 in 'LSK-myeloid' cluster). Compared to HSCs, baseline transcriptional heterogeneity in the LSK population was equally fluid but predominantly defined by an increased proportion of lineage-committed and mitotically active cells.

## Comparison of LSK progenitors identified HSC-specific responses to external signals

Analogous to HSCs we evaluated the effects of external stimulants on LSKs by both assessing changes in LSK distributions between clusters and differential gene expression within LSK clusters (*Figure 3C*, *E and G* and *Figure 3—figure supplement 2*, *Supplementary files 5-7 and 10*). Treatment with dmPGE$_2$ or poly(I:C) gave rise to novel clusters that were absent in control LSKs (*Figure 3B-D, G*). These treatment-induced LSK cell states displayed transcriptional profiles that were similar to the HSC equivalent cell states (*Figure 3A and E*). Poly(I:C) treatment induced two interferon responsive

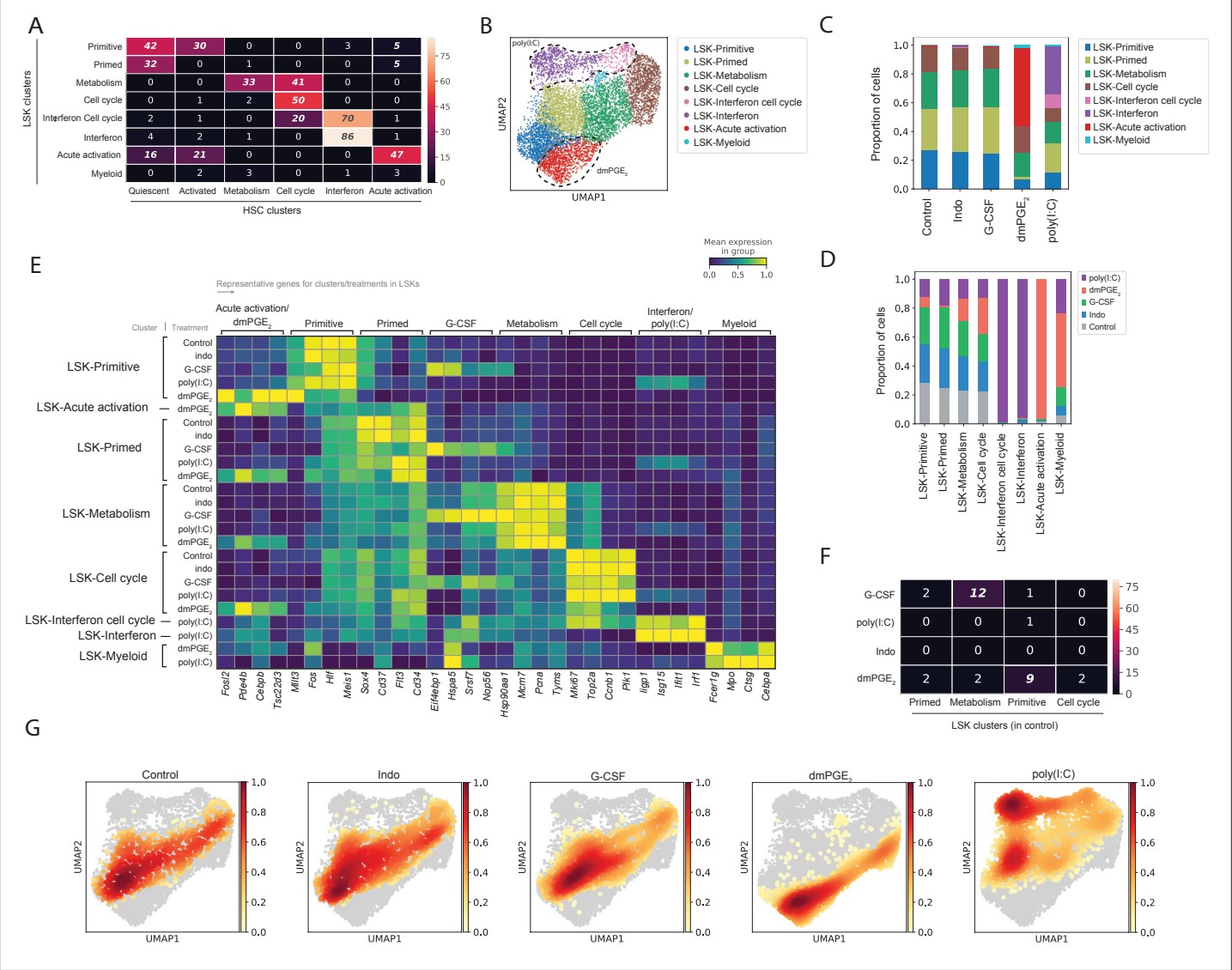

**Figure 3.** Comparative analysis of Lin-, c-Kit+, Sca1+ (LSK) response to external stimulants. (**A**) Heatmap with number of common genes between the 100 top enriched genes for LSK (rows) and hematopoietic stem cell (HSC) (columns) clusters, false discovery rate (FDR)-corrected hypergeometric p-values < 0.01 are italicized, exact p-values listed in *Figure 3—source data 1*. (**B**) Uniform manifold approximation and projection (UMAP) plot of LSK clustering (n = 8191 cells), with induced clusters by 16,16-dimethyl prostaglandin E$_2$ (dmPGE$_2$) (red) and poly(I:C) (pink and purple) traced with dashed line, see also *Figure 3—figure supplement 1*. (**C**) Proportion of LSK cells within clusters for each perturbation. (**D**) Proportion of LSK cells of each perturbation within a cluster normalized for total cell number per treatment. (**E**) Heatmap of selected enriched genes for each LSK cluster and treatment (columns, scaled expression) averaged gene expression for all cells within a cluster and treatment (rows, only clusters shown with >20 cells), see also *Figure 3—figure supplement 2*. (**F**) Heatmap with number of common genes between the 100 top induced genes per LSK treatment (rows) and LSK clusters (columns), FDR-corrected hypergeometric p-values < 0.01 are italicized, exact p-values listed in *Figure 3—source data 1*. (**G**) UMAP density graphs of LSK distribution for each external stimulant.

The online version of this article includes the following figure supplement(s) for figure 3:

**Source data 1.** Excel spreadsheet containing quantitative data for *Figure 3*.

**Figure supplement 1.** Multipotent progenitor (MPP) surface marker expression validates Lin-, c-Kit+, Sca1+ (LSK) cluster definitions.

**Figure supplement 2.** Heatmaps of differentially expressed genes in Lin-, c-Kit+, Sca1+ (LSKs) enable identification of genes and single-cell clusters with similar expression patterns.

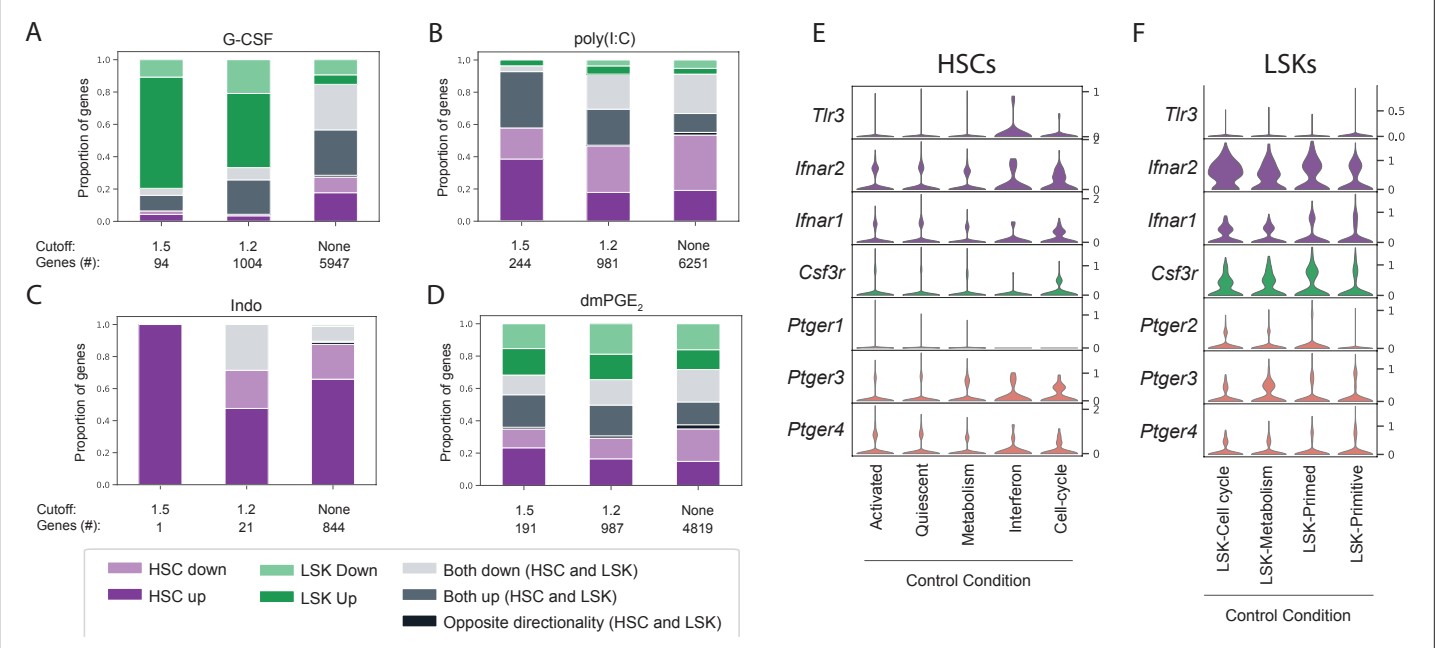

**Figure 4.** Lin-, c-Kit+, Sca1+ (LSK) and hematopoietic stem cell (HSC) cluster-specific differential gene expression cannot be explained by receptor expression. (**A–D**) Stacked bar graphs with proportion of differentially expressed genes that are unique for HSCs (purple), LSKs (green) or common (gray) upon granulocyte colony-stimulating factor (G-CSF) (**A**), poly(I:C) (**B**), Indo (**C**), or 16,16-dimethyl prostaglandin E$_2$ (dmPGE$_2$) (**D**) treatment. Below each bar graph the total number of differentially expressed genes ('genes #') for each fold-change ('cutoff') is listed. (**E–F**) Violin plots of receptor expression in control HSCs (**E**) and LSKs (**F**) split by cluster (only clusters with >20 cells displayed).

The online version of this article includes the following figure supplement(s) for figure 4:

**Source data 1.** Excel spreadsheet containing quantitative data for *Figure 4*.

clusters in LSKs, of which one showed higher mitotic activity ('LSK-interferon cell cycle', *Figure 3A, C and E*). Like in HSCs, G-CSF and indomethacin treatment did not alter cell proportions within LSK clusters (*Figure 3C and G*). In contrast to HSCs, in LSKs considerably less overlap existed between cluster-defining and stimulant-induced gene programs (*Figure 3F*). The poly(I:C)-induced gene program had no match to a baseline cluster identity because no interferon responsive cluster was present in unperturbed LSK cells (*Figure 3C and F*). For G-CSF a statistically significant but smaller (12%, p-value [hypergeometric test, FDR-corrected] = 10$^{-10}$) overlap existed between G-CSF-induced genes that were also 'LSK-metabolism' marker genes compared to HSCs (*Figure 3F*, *Supplementary file 9*). Overall, poly(I:C) and dmPGE$_2$ initiated a transcriptional program that altered the original LSK cell identity shifting cells between clusters. In contrast to HSCs, poly(I:C) induced the emergence of two new LSK cell clusters that did not exist in control. While responses to external stimuli were equally heterogeneous in the more differentiated LSK population, compared to HSCs, there was less crosstalk between LSK cell state heterogeneity at baseline and following perturbation of external signaling.

## Differential response to external signals in HSCs and LSK progenitors was not based on receptor expression

To evaluate and compare the magnitude of transcriptional changes in HSCs and LSKs in greater detail, DEGs for all four treatments at three levels of expression changes across all clusters, that is, using a 1.5-fold change, 1.2-fold change, and no fold-change cutoff (FDR < 0.01 see Materials and methods and *Supplementary file 7*) were compiled. We then aggregated genes based on common ('up/down overlap') or unique expression ('up/down HSC/LSK only') within HSCs or LSKs (*Figure 4A–D*). G-CSF perturbed gene expression more strongly within LSKs (green bars, *Figure 4A*) whereas stimulation by poly(I:C) predominantly affected HSCs (purple bars, *Figure 4B*). Receptor expression could not explain this difference since both the G-CSF receptor *Csf3r* and the type I interferon receptors *Ifnar1* and *Ifnar2* were expressed in a higher proportion of LSK cells compared to HSCs (*Figure 4E and F*). For perturbation of prostaglandin signaling indomethacin was found to selectively affect HSCs

(*Figure 4C*) whereas dmPGE$_2$ led to a balanced effect on HSCs and LSKs, with neither compartment dominating the DEGs (*Figure 4D*). In conclusion, different stimuli exhibited varying degrees of gene expression for either LSKs or HSCs. Receptor expression at baseline could not explain the variability of transcriptional responsiveness between HSCs and LSKs.

## HSC-specific chromatin architecture as potential cell intrinsic regulator of differential response to external signals

To better understand HSC intrinsic factors regulating the transcriptional 'receptiveness' to signals and resulting heterogeneous responses, we assessed chromatin states using scATAC-Seq (see Materials and methods) of sorted HSCs and MPPs. We clustered cells based on chromatin accessibility in HSCs resulting in two clusters ('HSC cluster 0' and 'HSC cluster 1', *Figure 5B*) and LSK cells consisting of MPPs and HSCs resulting in eight clusters (*Figure 5C* and *Figure 5—figure supplement 1A-B*, Materials and methods). To gain insight into the nature of the differentially accessible chromatin regions, we computed a per-cell transcription factor (TF) motif activity score using ChromVar (*Schep et al., 2017*) and evaluated enrichment of these scores across clusters. The motif activities of TFs CREB1, NF-κB, and STAT3 that are immediately downstream of prostaglandins, poly(I:C), and G-CSF (*Figure 5A*), respectively, were homogeneously distributed in HSCs (*Figure 5D*, *Figure 5—figure supplement 1C*) and the majority of LSK clusters (*Figure 5E*, *Figure 5—figure supplement 1D* and *Supplementary file 11*). This result suggested that HSCs have an equally responsive potential to these external signals based on their accessible chromatin states. We did detect differential enrichment of motifs for TFs that are further downstream in the response to external signals. Interferon regulatory factors (IRFs) that bind interferon signaling response elements (ISREs) are induced by NF-κB signaling as well as direct targets of poly(I:C) intracellular binding (*Negishi et al., 2018*, *Figure 5A*). The AP-1 motif can be bound by FOS and JUN, both are downstream effectors of the prostaglandin/CREB1 signaling pathway (*Luan et al., 2015*, *Figure 5A*). We found differential ISRE enrichment in HSC cluster 1 (log$_2$FC = 0.57, p-value(logistic regression) = 2.4*10$^{-5}$) and AP-1 enrichment in HSC cluster 0 (log$_2$FC = 2.6, p-value(logistic regression) = 3.0*10$^{-63}$, both indicated by asterisks, *Figure 5D* and *Figure 5—figure supplement 1C*). In addition, HSC cluster 0 displayed increased motif activity enrichment for several key HSC lineage-specific master TFs including RUNX (log$_2$FC = 1.3, p-value(logistic regression) = 8.0*10$^{-23}$) GATA (log$_2$FC = 0.68, p-value(logistic regression) = 7.2*10$^{-8}$), and Pu.1/SPI1 (log$_2$FC = 0.60, p-value(logistic regression) = 1.9*10$^{-9}$, indicated by asterisks, *Figure 5D* and *Figure 5—figure supplement 1C*) as well as SMAD, another signal-responsive TF (log$_2$FC = 0.87, p-value(logistic regression) = 2.7*10$^{-16}$, *Figure 5—figure supplement 1E, F*). In LSK cells the same motifs were also enriched in some clusters (top log$_2$FC indicated by asterisks, *Figure 5E* and *Figure 5—figure supplement 1D*, log$_2$FC and p-values in *Supplementary file 11*). However, no corresponding cluster like HSC cluster 0 existed where all lineage-specific (RUNX, GATA, and Pu.1) and signaling TF motifs (AP-1, SMAD) co-occured (*Figure 5E* and *Figure 5—figure supplement 1D*). In summary, the chromatin state directly downstream of external stimulants could not explain variability in gene expression upon treatment in HSCs. Rather, our analysis implicated cell intrinsic heterogeneity of downstream effectors, such as AP-1 and IRFs that may govern differential transcriptional responses. While cluster enrichment of AP-1 and ISREs was not unique to HSCs, we observed a specific co-occurrence of AP-1 and HSC lineage-specific master factors suggestive of HSC unique chromatin architecture.

## Discussion

Here, we provide a comprehensive transcriptional and epigenetic single cell analysis of a highly purified, functionally validated HSC population. Our work reveals that HSCs exist in fluent transcriptional and epigenetic states rather than distinctly separated cell types. While we cannot entirely rule out that the continuous cell states arose from the noisy nature of scRNA-Seq sampling, this is unlikely given our observation that genes that vary along the same transcriptional gradients are also functionally correlated (e.g. IEGs). External perturbations rapidly shifted HSC distribution between HSC states within hours of signaling, providing evidence that the transcriptional states are highly dynamic allowing HSCs to quickly transition between states. Interestingly, we observed heterogeneity of HSC responses to external stimuli which may be determined by the baseline transcriptional and epigenetic state supported by our single-cell chromatin studies. Preliminary findings suggested an HSC specific

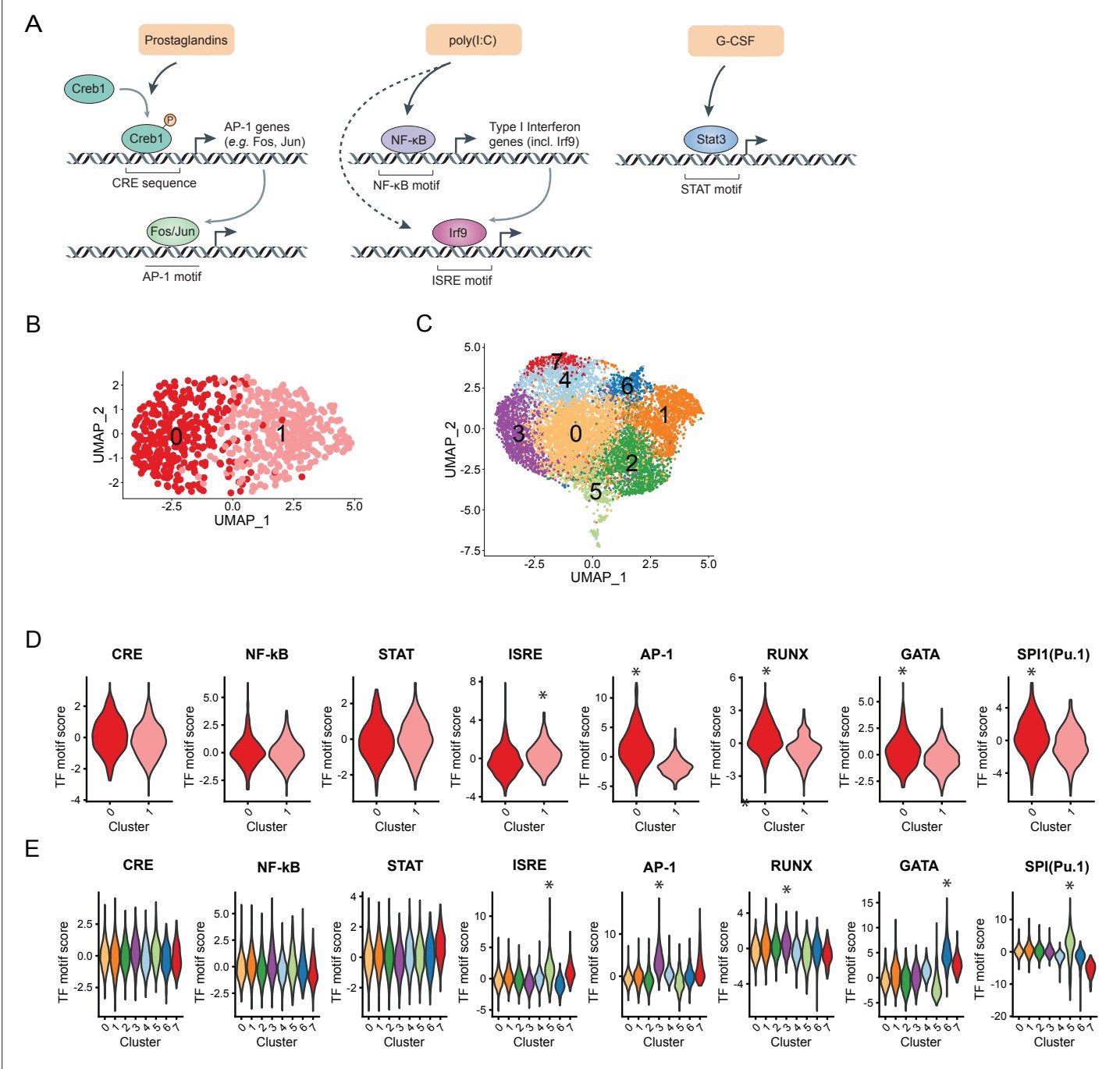

**Figure 5.** Heterogeneous distribution of interferon signaling response element (ISRE) and AP-1 motif in hematopoietic stem cells (HSCs) and Lin-, c-Kit+, Sca1+ (LSKs) and specific motif co-occurrences in HSCs. (**A**) Schematic of downstream transcriptional signaling pathways for external stimulants. (**B–C**) Uniform manifold approximation and projection (UMAP) plot of HSC (**B**) single-cell chromatin accessibility sequencing (scATAC-Seq) clusters (n = 730 cells) or LSK (**C**) scATAC-Seq clusters (n = 10,750 cells), see also *Figure 5—figure supplement 1A-B*. (**D–E**) Violin plots of transcription factor (TF) motif scores enriched in HSCs (**D**) and LSKs (**E**) with selected significant p-values (logistic regression) indicated by asterisks, see also *Supplementary file 11* and *Figure 5—figure supplement 1C-D*.

The online version of this article includes the following figure supplement(s) for figure 5:

**Figure supplement 1.** Uniform distribution of motif activity immediately downstream of external stimulants and differential enrichment for secondary signals in hematopoietic stem cells (HSCs) and Lin-, c-Kit+, Sca1+ (LSKs).

co-occurrence of signaling and lineage-specific TF motif activities that is consistent with previous

observations in human hematopoietic progenitors (*Trompouki et al., 2011*; *Choudhuri et al., 2020*). Overall, our data indicates that the single-cell landscape of in vivo-derived, functional HSCs is likely made up of a unique chromatin architecture with fluent transcriptional states, some of which can be rapidly influenced by external signals.

Our combined scRNA-Seq and cell hashing (HTO barcoding) approach allowed us to gain insights into the transcriptional landscape of HSCs and phenotypically defined MPP populations within the LSK compartment at steady state and following perturbations with extrinsic signals. Our results enabled us to connect the transcriptional profile on a single-cell level to the previously described phenotypic behaviors of these MPP populations (; *Pietras et al., 2015*; *Cabezas-Wallscheid et al., 2014*). For example even though both MPP2 and MPP3 cells have been previously described as myeloid biased (*Pietras et al., 2015*), our analysis allowed to determine the proportion of putative myeloid cells within MPP2 and MPP3/4 cells as well as the relative MPP2 and MPP3/4 composition of myeloid cells. The HTO barcoding method provided a flexible tool to evaluate and compare transcriptional profiles within phenotypically defined populations because the technology used here is not dependent on the availability of specifically conjugated antibodies against particular surface receptors. In addition, *Xist* expression was used to deconvolute pooled male and female cells. While our analysis revealed only minimal sexual dimorphism that is consistent with previous reports (*Nakada et al., 2014*; *Gal-Oz et al., 2019*), the negligible additional investment to obtain data from both sexes may become the default experimental design in mammalian scRNA-Seq experiments. Our work presents evidence for two value-adding pooling strategies that allow for further insights into cell populations analyzed by scRNA-Seq.

We used a two-pronged strategy to assess the specificity of external perturbations in HSCs and LSKs. First, we determined changes of cell proportions between cell states. Second, we evaluated differential expression within particular cell states following stimulation. Comparison of cluster-enriched and treatment-induced genes allowed us to identify unique and common genes for a given perturbation or a specific cluster. In contrast to LSKs, HSCs exhibited a high degree of overlap between stimulant-induced and cluster marker-defined gene programs. These results suggest that even at baseline, HSC transcriptional heterogeneity is defined by differences in signaling activity. Changes in cell proportions between different clusters indicated further specificity for treatment and differentiation state. Poly(I:C) and dmPGE$_2$ led to cellular shifts between distinct transcriptional states with poly(I:C) driving the formation of two novel interferon-related clusters in LSKs but not HSCs. The strength of transcriptional perturbation could not solely be estimated based on the distribution of cells within clusters alone. G-CSF did not change the cell proportions between clusters but rather elicited strong transcriptional responses within a given cell state. Comparison of DEGs within clusters in HSCs and LSKs indicated that HSCs display a smaller response across all clusters to G-CSF compared to LSK progenitors. In summary, scRNA-Seq enabled a number of analyses that uncovered novel, HSC-specific responses to external perturbations.

We evaluated the effect of three complementary signaling pathways (G-CSF, prostaglandin, and interferon) on the transcriptional state of HSCs. Pharmacological perturbation of these signaling pathways allowed to tightly control critical experimental parameters (e.g. genetic background of mice, timing of sample processing) that mitigated potential confounders of the downstream analysis. With the exception of indomethacin, we chose a short treatment window of 2 hr to increase the likelihood of studying direct downstream effects of stimulants on HSCs. Analysis of DEGs within clusters indicated interferon- vs. Toll-like receptor response genes induced by poly(I:C) treatment. While we could not detect transcripts for Type I interferons in our scRNA-Seq data of HSCs or MPPs, it is possible that some of the interferon-response genes were induced indirectly by release of interferons from the niche. An interferon inducer similar to poly(I:C) has been previously shown to increase IFNα protein levels in the serum as early as 2 hr post in vivo injection (*Linehan et al., 2018*). Future work using genetic models is needed to further dissect indirect vs. direct effects of external stimulants on HSCs.

There is a tradeoff between the strength of a perturbation required for experimental robustness vs. studying signals that are more physiologically relevant but lead to more subtle changes within and between cells. Here, we evaluated response of HSCs to three different external activators mimicking niche signals that were dosed two to four orders of magnitude higher than what an animal would typically encounter during actual injury or infection (*Eyles et al., 2008*; *Porter et al., 2013*; *Hoggatt et al., 2013*; *Sheehan et al., 2015*). To assess niche-derived signals in a more physiological setting, we

administered the Cox1/2 inhibitor indomethacin orally for 1 week to deplete endogenous prostaglandins. As expected, the changes in gene expression with indomethacin were much weaker than those observed after acute injection with dmPGE$_2$, G-CSF, and poly(I:C). ScRNA-Seq analysis offers unique tools to evaluate gene expression changes in response to weak perturbations. Pseudotime analysis showed that depletion of endogenous prostaglandins using indomethacin led to a small but significant shift in the transcriptional state of HSCs. The effect of indomethacin on IEGs such as *Fos* was further validated in independent FACS experiments which showed that the transcriptional programs implicated through pseudotime were also found to be perturbed using this orthogonal assay. How exactly the increase in RNA levels of *Fos* observed in scRNA-Seq can be reconciled with decreased FOS protein levels determined by FACS analysis will need to be addressed in future experiments. Another important implication and potential caveat highlighted by our findings is that RNA and protein levels may not always positively correlate, even on a single-cell level. Regardless, scRNA-Seq technologies provide sensitive tools to interrogate subtle changes in cellular states.

In summary, we showed that single-cell approaches provide a rich and sensitive tool to analyze transcriptional and epigenetic states of HSCs during homeostasis and upon external perturbation. We found that HSCs exist in dynamic cell states and external signals can induce rapid transitions between, as well as changes within, these HSC states. While our work did not reveal whether these transcriptional states are associated with specific niches in vivo, novel spatial transcriptomic approaches provide exciting new opportunities to address such questions (*Rodriques et al., 2019*). Additionally, recently developed barcoding strategies enable assessment of treatment-induced transcriptional changes and functional potential of single cells within the same experiment (*Rodriguez-Fraticelli et al., 2020*). Understanding endogenous levels of niche-derived factors and the associated transcriptional and epigenetic responses will advance our basic understanding of stem cells and their potential applications in the clinic.

# Materials and methods

## Key resources table

| Reagent type (species) or resource | Designation | Source or reference | Identifiers | Additional information |
|---|---|---|---|---|
| Genetic reagent (*Mus musculus*) Male and female | Replicate 1 | Jackson Laboratory | RRID:IMSR_JAX:016617 | |
| Genetic reagent (*Mus musculus*) Male and female | Replicate 2, CD45.2 (transplant recipients) | Jackson Laboratory | RRID:IMSR_JAX:000664 | Used for pharmacological perturbations |
| Genetic reagent (*Mus musculus*) Female only | Transplant donors | Jackson Laboratory | RRID:IMSR_JAX:002014 | |
| Antibody | Anti-CD117 (c-Kit), ACK2, APC (rat monoclonal) | Thermo Fisher Scientific (17-1172-83) | RRID:AB_469434 | FACS (1:100) |
| Antibody | Anti-CD11b/Mac1, M1/70, eFluor 450 (rat monoclonal) | Thermo Fisher Scientific (48-0112-80) | RRID:AB_1582237 | FACS (1:100) |
| Antibody | Anti-CD11b/Mac1, M1/70, PE-Cyanine5 (rat monoclonal) | Thermo Fisher Scientific (15-0112-83) | RRID:AB_468715 | FACS (1:100) |
| Antibody | Anti-CD11b/Mac1, M1/70, Alexa Fluor 700 (rat monoclonal) | BD Pharmingen (557960) | RRID:AB_396960 | FACS (1:300) |
| Antibody | Anti-CD135 (Flt3), A2F10, PE (rat monoclonal) | Thermo Fisher Scientific (12-1351-81) | RRID:AB_465858 | FACS (1:100) |
| Antibody | Anti-CD150, TC15-12F12.2, PE/Cy7 (rat monoclonal) | Biolegend (115914) | RRID:AB_439797 | FACS (1:100) |

*Continued on next page*

*Continued*

| Reagent type (species) or resource | Designation | Source or reference | Identifiers | Additional information |
|---|---|---|---|---|
| Antibody | Anti-CD3, 17A2, APC (rat monoclonal) | Thermo Fisher Scientific (17-0032-82) | RRID:AB_10597589 | FACS (1:100) |
| Antibody | Anti-CD34, RAM34, eFluor 450 (rat monoclonal) | Thermo Fisher Scientific (48-0341-80) | RRID:AB_2043838 | FACS (1:33) |
| Antibody | Anti-CD34, RAM34, FITC (rat monoclonal) | Thermo Fisher Scientific (11-0341-85) | RRID:AB_465022 | FACS (1:33) |
| Antibody | Anti-CD3e, 145–2C11, eFluor 450 (armenian hamster monoclonal) | Thermo Fisher Scientific (48-0031-80) | RRID:AB_10733280 | FACS (1:100) |
| Antibody | Anti-CD3e, 145–2C11, PE-Cyanine5 (armenian hamster monoclonal) | Thermo Fisher Scientific (15-0031-83) | RRID:AB_468691 | FACS (1:100) |
| Antibody | Anti-CD45.1, A20, FITC (mouse monoclonal) | BD Pharmingen (553775) | RRID:AB_395043 | FACS (1:100) |
| Antibody | Anti-CD45.2, 104, PE (mouse monoclonal) | BD Pharmingen (560695) | RRID:AB_1727493 | FACS (1:100) |
| Antibody | Anti-CD45R (B220), RA3-6B2, eFluor 450 (rat monoclonal) | Thermo Fisher Scientific (48-0452-80) | RRID:AB_1548763 | FACS (1:100) |
| Antibody | Anti-CD45R (B220), RA3-6B2, PE-Cyanine5 (rat monoclonal) | Thermo Fisher Scientific (15-0452-83) | RRID:AB_468756 | FACS (1:100) |
| Antibody | Anti-CD45R/ (B220), RA3-6B2, pacific Blue (rat monoclonal) | Biolegend (103227) | RRID:AB_492876 | FACS (1:100) |
| Antibody | Anti-CD48, HM48-1, Alexa Fluor 700 (armenian hamster monoclonal) | Biolegend (103425) | RRID:AB_10612754 | FACS (1:100) |
| Antibody | Anti-CD5, 53–7.3, eFluor 450 (rat monoclonal) | Thermo Fisher Scientific (48-0051-80) | RRID:AB_1603252 | FACS (1:100) |
| Antibody | Anti-CD8a, 53–6.7, eFluor 450 (rat monoclonal) | Thermo Fisher Scientific (48-0081-80) | RRID:AB_1272235 | FACS (1:100) |
| Antibody | Anti-c-Fos, H15-S, FITC (rabbit monoclonal) | Abcam (ab175647) | RRID:AB_2893164 | FACS (10 µl for 1 MIO cells) |
| Antibody | Anti-Ly-6A/E (Sca-1), D7, PE-eFluor 610 (rat monoclonal) | Thermo Fisher Scientific (61-5981-80) | RRID:AB_2574647 | FACS (1:100) |
| Antibody | Anti-Ly-6A/E (Sca-1), D7, APC/Cy7 (rat monoclonal) | Biolegend (108125) | RRID:AB_10639725 | FACS (1:100) |
| Antibody | Anti-Ly-6G (Gr-1), RB6-8C5, eFluor 450 (rat monoclonal) | Thermo Fisher Scientific (48-5931-80) | RRID:AB_1548797 | FACS (1:100) |

*Continued on next page*

*Continued*

| Reagent type (species) or resource | Designation | Source or reference | Identifiers | Additional information |
|---|---|---|---|---|
| Antibody | Anti-Ly-6G (Gr-1), RB6-8C5, PE-Cyanine5 (rat monoclonal) | Thermo Fisher Scientific (15-5931-83) | RRID:AB_468814 | FACS (1:100) |
| Antibody | Anti-Ly-6G (Gr-1), RB6-8C5, PE-Cyanine7 (rat monoclonal) | Thermo Fisher Scientific (25-5931-82) | RRID:AB_469663 | FACS (1:100) |
| Antibody | Anti-TER-119/ Erythroid Cells, TER-119, eFluor 450 (rat monoclonal) | Thermo Fisher Scientific (48-5921-80) | RRID:AB_1518809 | FACS (1:100) |
| Antibody | Anti-TER-119/ Erythroid Cells, TER-119, PE-Cyanine5 (rat monoclonal) | Thermo Fisher Scientific (15-5921-83) | RRID:AB_468811 | FACS (1:100) |
| Antibody | Anti-TER-119/ Erythroid Cells, TER-119, APC/Cy7 (rat monoclonal) | Biolegend (116223) | RRID:AB_2137788 | FACS (1:100) |
| Antibody | TotalSeq-A0301 anti-mouse Hashtag 1 Antibody, M1/42; 30-F11 (rat monoclonal) | Biolegend (155801) | RRID:AB_2750032 | Cell hashing (1 µg per reaction) |
| Antibody | TotalSeq-A0302 anti-mouse Hashtag 2 Antibody, M1/42; 30-F12 (rat monoclonal) | Biolegend (155803) | RRID:AB_2750033 | Cell hashing (1 µg per reaction) |
| Antibody | TotalSeq-A0303 anti-mouse Hashtag 3 Antibody, M1/42; 30-F13 (rat monoclonal) | Biolegend (155805) | RRID:AB_2750034 | Cell hashing (1 µg per reaction) |
| Antibody | TotalSeq-A0304 anti-mouse Hashtag 4 Antibody, M1/42; 30-F14 (rat monoclonal) | Biolegend (155807) | RRID:AB_2750035 | Cell hashing (1 µg per reaction) |
| Reagent, commercial | Streptavidin, -, PE-Cyanine5 | Thermo Fisher (15-4317-82) | RRID:AB_10116415 | FACS (1:100) |
| Reagent, commercial | Streptavidin, -, eFluor 450 | Thermo Fisher Scientific (48-4317-82) | RRID:AB_10359737 | FACS (1:100) |
| Commercial assay or kit | scRNA-Seq kit V2 – replicate 1 | 10× Genomics | PN-120267 | |
| Commercial assay or kit | scRNA-Seq kit V3– replicate 2 | 10× Genomics | PN-1000075 | |
| Commercial assay or kit | scATAC-Seq kit | 10× Genomics | PN-1000111 | |
| Commercial assay or kit | Lineage depletion kit | Miltenyi Biotech | 130-090-858 | |
| Chemical compound, drug | Poly(I:C) HMW | Invivogen | tlrl-pic-5 | |
| Chemical compound, drug | DmPGE2 | Cayman | 14750 | |

*Continued on next page*

*Continued*

| Reagent type (species) or resource | Designation | Source or reference | Identifiers | Additional information |
|---|---|---|---|---|
| Chemical compound, drug | G-CSF | Thermo Fisher | PHC2031 | |
| Chemical compound, drug | Indomethacin | Sigma | PHR1247-500MG | |
| Software, algorithm | GraphPad Prism | GraphPad (Version 6.05) | RRID:SCR_002798 | https://www.graphpad.com/ |
| Software, algorithm | FlowJo (Tree Star) | FlowJo (Version 10.5.3) | RRID:SCR_008520 | https://www.flowjo.com/ |
| Software, algorithm | Cellranger | 10× Genomics | v3.0.1 v2.1.0 (Replicate 1) v1.2.0 (scATAC-Seq) | https://support.10xgenomics.com/single-cell-gene-expression/software/overview/welcome |
| Software, algorithm | CITE-Seq count | https://hoohm.github.io/CITE-seq-Count/ (version 1.4.3) | RRID:SCR_019239 | https://github.com/Hoohm/CITE-seq-Count, *Roelli, 2021* |
| Software, algorithm | Scanpy | (*Wolf et al., 2018*) Various versions, see jupyter notebooks + dockerhub for documentation | RRID:SCR_018139 | https://scanpy.readthedocs.io/en/stable/ |
| Software, algorithm | pegasuspy | *Gaublomme et al., 2019* | Version 0.17.1 | https://github.com/klarman-cell-observatory/pegasus/tree/0.17.1, *Yang, 2021* |
| Software, algorithm | Signac | (*Stuart et al., 2019*) Version 0.2.5 | RRID:SCR_021158 | https://satijalab.org/signac/ |
| Software, algorithm | GitHub | This paper | https://github.com/evafast/scrnaseq_paper, copy archived at swh:1:rev:231286dc1447516f938bed8191839edb554a4fd3 (*Fast, 2021*) | Code for all analyses + description |
| Software, algorithm | Dockerhub | This paper | https://hub.docker.com/u/evafast1 | Docker images for analysis |
| Software, algorithm | UCSC cell browser | *Speir et al., 2021* | https://cells.ucsc.edu/ | Interactive app |

## Wet lab methods

### Mice and external stimulant treatment

For the HSC Replicate 1 experiment, we used the following mouse strain (#016617) that was obtained from Jackson labs but bred in-house. For external stimulant treatments, male and female mice (8–10 weeks) were ordered from Jackson labs (strain CD 45.2 [Ly5.2], #00664). Mice were kept for at least 1 week in the animal facility before initiating experiments and allocated at random (by cage) into experimental groups. Indomethacin (Sigma, 6 mg/l) was administered for 7 days in acidified drinking water to maintain stability (*Curry et al., 1982*; *Praticò et al., 2001*). Indomethacin supplemented drinking water was changed every other day. Mice were injected with the following drugs and euthanized after 2 hr: poly(I:C) HMW (Invivogen), IP injection 10 mg/kg (*Pietras et al., 2014*). G-CSF Recombinant Human Protein (Thermo Fisher), IP injection, 0.25 mg/kg (*Morrison et al., 1997*). dmPGE$_2$ (Cayman), SC injection, 2 mg/kg (*Hoggatt et al., 2013*). Mice were weighed before injection and injection volume was adjusted to ensure equal dose between individual mice. The 'control' condition from the external stimulant treatments was also used as the second independent biological replicate of unperturbed HSCs (HSC Replicate 2). All animal procedures were approved by the Harvard University Institutional Animal Care and Use Committee.

## Bone marrow preparation and FACS

Whole bone marrow was isolated from femur, tibia, hip, and vertebrae via gentle crushing using a mortar and pestle. Stem and progenitor cells were enriched via lineage depletions (Miltenyi Biotech, 130-090-858). Antibodies, dilutions, and vendors are listed in the Key resources table. Cells were stained for 1.5 hr based on published best practice protocols for assessing CD34 labeling (*Ema et al., 2006*). HSCs (LSK, CD48-, CD150+, CD34-), MPP1s (LSK, CD48-, CD150+, CD34+), MPP0s (LSK, CD48-, CD150-), MPP2s (LSK, CD48+, CD150+), and MPP3/4s (LSK, CD48+, CD150-) were sorted on a FACSAria (Becton Dickinson) and representative sorting scheme is shown in *Figure 1—figure supplement 1A*. Purity of >80% was ensured by reanalyzing each sorted population.

## Sample size estimation and sample batching

To determine appropriate sample sizes of mice and HSCs, we performed an initial experiment on fresh HSCs (HSC Replicate 1) which yielded estimated number of 2382 cells (after filtering), and which resolved biologically meaningful clusters (*Figure 1—figure supplement 2A*). In subsequent experiments we therefore targeted obtaining a similar or higher cell number. For external stimulant treatment, we based our sample size of five male and five female mice on this initial experiment. Because of sample processing times, a maximum of two conditions could be performed on the same day, resulting in three separate days of experiments. To mitigate batch effects resulting from different experimental days, the following precautions were taken. (1) All mice included in the external stimulant treatment were ordered from the same batch from JAX. (2) Control mice were administered acidified water and injected with DMSO to control for both unspecific perturbations that might result from the external stimulant treatments. (3) All experiments were performed within less than 1 week and single-cell libraries were prepared together for all samples after the initial droplet reaction was frozen. (4) FACS gates were set up initially but left constant for each experiment. Single color controls as well as fluorescence minus one controls ensured that there was minimal day-to-day technical drift on the FACS instrument.

## Intracellular staining for FACS

BM extraction, lineage depletion, and surface marker staining were performed as described above. Cells were fixed and permeabilized for intracellular staining according to manufacturer's instructions (BD Biosciences, 554714). Intracellular staining was performed for 30 min on ice. Samples were analyzed on an LSRII FACS analyzer.

## Limiting dilution transplantation assay

Recipient CD45.2 (Jax #00664) mice were gamma-irradiated (Cs-137 source) with a split dose of 5.5 Gy each 1 day before transplantation. HSCs were isolated from CD45.1 (Jax #002014) donors and transplanted with 200,000 whole bone marrow cells (CD45.2) via retro-orbital injection. Donor cell engraftment was monitored monthly for 16 weeks using an LSRII FACS analyzer (Becton Dickinson). Flow cytometry data were analyzed with FlowJo (Tree Star). HSC frequency was calculated using the following website: http://bioinf.wehi.edu.au/software/elda/.

## Single-cell RNA and ATAC sequencing library preparation and sequencing

Male and female cells were sorted separately but pooled in equal ratios before further downstream processing. For CITE-Seq HTO labeling of MPP populations, 0.25 µg of TruStain FcX Blocking reagent (Biolegend) was added for 10 min on ice. Each MPP populations was labeled with 1 µg of TotalSeq antibody cocktail (Biolegend, see Key resources table) and incubated for 30 min on ice. After washing, cells were resuspended in small amounts, counted and pooled in equal ratios. Each drug treatment condition resulted in one pooled MPP and one HSC sample that were processed separately for scRNA-Seq according to manufacturer's recommendations (10× Genomics, 3' V2 for HSC Replicate 1 experiment and V3 for external stimulant treatments). Briefly, for pooled MPPs, no more than 10,000 cells were loaded. For HSCs, all sorted cells (between 2222 sorted events for dmPGE$_2$ and 12,017 sorted events for control) were loaded on the 3' library chip. For preparation of HTO – surface libraries manufacturer's recommendations (Biolegend) were followed. For ATAC-Seq, HSCs and MPPs (pooled MPP0, MPP1, MPP2, and MPP3/4) were sorted as described above from five male

and five female mice (strain CD 45.2 [Ly5.2], JAX strain #00664). Nuclei were isolated and libraries were prepared using manufacturer's recommendations (10× Chromium Single Cell ATAC). Libraries were sequenced on a Next-seq 500, 75 cycle kit ('Replicate 1', scRNA-Seq) and NOVAseq 6000, 100 cycle kit ('Replicate 2' and external stimulant treatments, scRNA-Seq, scATAC-Seq).

## Computational and statistical analyses

All code and a detailed description of the analysis is available in a dedicated GitHub repository (see link in key resources table). To ensure reproducibility the entire analysis (except cellranger and CITE-Seq count) was entirely performed in Docker containers. Containers used for the analysis are indicated in the Jupyter notebooks and corresponding images are available on dockerhub (see link in key resources table). Interactive cell browser web app is available here: (https://mouse-hsc.cells.ucsc.edu). Raw data are available with GEO accession code GSE165844.

## Demultiplexing and generation of count matrices

Cellranger (v3.0.1) command 'mkfastq' was used to demutliplex raw base call (BCL) files into individual samples and separate mRNA FASTQ files and HTO surface fastq files. The cellranger 'count' command was used with default options to generate gene by cell matrices from mRNA FASTQ files. CITE-Seq count (version 1.4.3) was used to generate surface count by cell matrices from the HTO surface FASTQ libraries. For the fresh HSC Replicate 1 experiment cellranger (version 2.1.0) was used for demultiplexing and count matrix generation. The mm10 reference genome was used for all alignments. For scATAC-Seq cellranger-atac mkfastq and count (1.2.0) was used for demultiplexing and alignment and generation of the fragment file. To generate the count matrix MACS2 was run with default parameters (keeping duplicates) on the aligned reads. Resulting peak summits were extended to 300 bp and counts were extracted from fragment file using a custom script (see GitHub repository) to generate a count matrix.

## Quality control, filtering, and dimensionality reduction of scRNA-Seq data

The main parts of the bioinformatic analysis of scRNA-Seq data was performed using the python package scanpy (*Wolf et al., 2018*). For filtering and quality control, best practice examples were followed (*Luecken and Theis, 2019*). Count matrices were filtered on a gene and cell level. Cells were excluded with either less than 3000 UMIs, less than 1500 (LT), or 2000 (MPPs) genes or more than 20,000 (LT) or 30,000 (MPPs) counts. A cutoff of no more than 10% UMIs aligned to mitochondrial genes per cell was applied. Genes expressed in less than 20 cells were excluded from the analysis. Counts were normalized to 10,000 per cell and log transformed. Features (genes) were scaled to unit variance and zero mean before dimensionality reduction. To reveal the structure in the data, we built a neighborhood graph and used the leiden community detection algorithm (*Traag et al., 2019*) to identify communities or clusters of related cells (see also below). The UMAP algorithm was used to embed the high-dimensional dataset in a low-dimensional space (*Becht et al., 2018*). DPA was used for comparing cell proportions between clusters as previously described (*Farbehi et al., 2019*). Interactive visualization app of scRNA-Seq data was prepared using UCSC Cell Browser package (*Speir et al., 2021*).

## Demultiplexing of CITE-Seq hashtag data

We used the DemuxEM (*Gaublomme et al., 2019*) implementation in pegasuspy to assign MPP surface identities and demultiplex the pooled MPP sample. First background probabilities ('pg.estimate_background_probs') were estimated using default settings and 'pg.demultiplex' was run adjusting the alpha and the alpha_noise parameter to maximize cell retrieval by singlet classification. Assignments were validated by plotting count matrix in UMAP space and observing four distinct clusters indicative for the four HTO labels that were pooled. The proportion of demultiplexed cells matched the original pooling ratio. Analysis of coexpression of sex-specific genes allowed for further validation of the doublet rate. Proportion of cells classified by DemuxEM as doublets exceeded doublet rate estimated by coexpression of sex-specific genes.

## Batch correction

Because of timing required for FACS and sample prep, it was impossible to obtain HSCs and MPPs from all conditions on 1 day (see also 'Sample size estimation and sample batching' above). To evaluate if batch correction was needed, we determined scRNA-Seq clusters and enriched genes by processing each sample separately or by combined analysis of all samples. Even though similar scRNA-Seq clusters were found in individual samples, these populations were non-overlapping in the integrative analysis (especially for G-CSF). To correct for the batch effects we used ComBat (*Johnson et al., 2007*) with default settings on the $\log_2$ expression matrix, allowing cells to be clustered by cell type or cell state. Batch correction results were similar when we used Scanorama (*Hie et al., 2019*) and Harmony (*Korsunsky et al., 2019*) but both of these methods appeared to be overcorrecting with respect to the dmPGE$_2$-treated population. To correct for potential sex-specific differences Xist counts were regressed out. Raw data was used for all differential expression analyses and plotting of single-cell gene expression values. Batch-corrected counts were used for clustering and DPT analysis.

## Optimal cluster parameter selection

Since HSCs and MPPs are highly purified cell populations, we did not observe any clearly separated clusters in UMAP space. To aid the optimal choice of hyperparameters for leiden clustering, we used a combination of Silhouette coefficient and Davies–Bouldin index. We first validated this approach using the PBMC3K (from 10× genomics, scanpy.datasets.pbmc3k()) silver standard dataset. We iterated through a range of KNN nearest neighbors and Leiden resolution combinations measuring average Silhouette coefficient and Davies–Bouldin index in PCA space for each combination. Plotting the optimal value for Silhouette score and Davies–Bouldin index vs. increasing numbers of clusters allowed for the determination of appropriate cluster number for the dataset. For the PBMC dataset, there was a clear drop-off in optimal value after eight clusters, which is corroborated by most single-cell tutorials that also report eight clusters for this dataset. After validation of this approach on PBMCs, we assessed Silhouette coefficient and Davies–Bouldin index for different clustering results of our own HSC and MPP datasets. This allowed us to select the optimal hyperparameters for each cluster number. The approach was validated by comparing two independent biological replicates of control HSCs ('Replicate 1' and 'Replicate 2').

## Differential expression using MAST

Differential expression analysis was performed using MAST (*Finak et al., 2015*). This method is based on a Hurdle model that takes into account both the proportion of cells expressing a given transcript and transcript levels themselves while being able to control for covariates. Based on previous reports, differential expression cutoff was set at 1.2-fold (*Smillie et al., 2019*) and a more stringent cutoff of 1.5-fold was also included. Only genes that were expressed in at least 5% of the cells were considered for differential expression analysis. FDR (Benjamini and Hochberg) cutoff was set at 1%. For drug treatments, differential expression between treatment and control was assessed within the entire LSK or HSC dataset and within each cluster controlling for number of genes per cell and sex. For differential expression analysis between male and female cells at baseline, control datasets were analyzed with clusters and number of genes as a covariates. For sex-specific effects of drug treatments, samples were split by sex and analyzed separately. Resulting differential expression coefficients were compared between male and female cells. To identify gene signatures with common patterns, for each treatment average expression of DEGs was extracted per cluster, scaled (z-score) and grouped together by similarity using hierarchical clustering (seaborn.clustermap, Euclidean distance, single linkage).

## DPT analysis

For DPT analysis (*Haghverdi et al., 2016*), cells from the 'Quiescent' and 'Activated' cluster were selected for the following treatments: control, indomethacin, and G-CSF. We recalculated PCA and UMAP embeddings in this reduced dataset. Re-clustering using the Leiden algorithm was used to exclude outlier cells and assess top enriched genes within the new 'Activated' cluster. Raw expression of the three top enriched genes (Nr4a1, Nr4a2, Hes1) was summed to robustly select the most highly 'Activated' cell as a root cell. DPT was calculated with the following function in scanpy ('sc.tl. dpt') using default settings. Cells were ranked according to pseudotime and kernel density distribution was plotted using a bandwidth of 0.02. The Mann–Whitney U-test was used to assess if cells

from different samples are drawn from the same pseudotime distribution. To analyze gene expression across pseudotime, for each sample cells were split into 10 equally sized bins according to ascending pseudotime. Bin 1 contained the first 10% of cells with the lowest pseudotime and bin 10 contained the 10% of cells with the highest pseudotime. Average gene expression for representative genes were plotted for each bin and sample.

### Pathway and gene list enrichment analysis and comparison

We performed over-representation analysis comparing various gene sets of interest (upregulated by stimulants, enriched in clusters) to a reference gene set. Depending on the analysis, the reference gene set was composed of an entire database of pathways (REACTOME, GO:BP), manually curated pathways of interest (searching for keywords on MSigDB database and from relevant publications; *Goessling et al., 2011*; *Schuettpelz et al., 2014*; *Pedersen et al., 2016*; *Giladi et al., 2018*; *Mervosh et al., 2018*; *Patterson et al., 2020*; *Cilenti et al., 2021*; *Rodriguez-Fraticelli et al., 2020*; *Cabezas-Wallscheid et al., 2017*) or gene sets generated from the analysis itself (marker genes from other clusters). Enrichment was assessed using a hypergeometric test (one-sided Fisher's exact test) and p-values were corrected for FDR using Benjamini–Hochberg. We deliberately choose to evaluate the top 100 genes for every pairwise cluster/treatment comparisons to be more intuitive to interpret and compare.

### Calculation of transcriptional scores

Transcriptional scores for each cluster were calculated using the scanpy function 'scanpy.tl.score_genes'. Briefly the score represents the average expression of a set of genes subtracted with the average expression of a reference set of genes. The reference set is randomly sampled for each binned expression value. Mean scores per cluster were compared via ANOVA followed by Tukey's HSD test for individual post hoc mean comparisons.

### scATAC-Seq

The R package Signac (version 0.2.5), an extension of Seurat (*Stuart et al., 2019*), was used for quality control, filtering of ATAC-Seq peaks counts and plotting. Quality of scATAC-Seq dataset was ensured by presence of nucleosomal banding pattern and enrichment of reads around transcription start sites. Cells were removed with a less than 1000 or more than 20,000 fragments in peaks. Male and female cells were classified according to absence or presence of Y-chromosome reads. Since distribution of male and female cells appeared uniform across all analyses, no downstream correction was taken for sex. Term frequency-inverse document frequency was used for normalization and dimensionality reduction was performed by singular value decomposition. Cells were clustered using the Louvain community finding algorithm after a neighborhood graph was built with k = 20 (HSCs) or k = 30 (LSK) nearest neighbors. To calculate TF motif scores, ChromVAR (*Schep et al., 2017*) was run with default parameters using the JASPAR 2018 motif database. Differential TF motif activity scores between clusters were calculated with the 'FindMarkers' function in Signac using a logistic regression and p-values were adjusted using a Bonferroni correction.

## Acknowledgements

The authors thank members of the Zon, Wagers, Camargo, and Scadden lab for helpful technical and scientific discussions, Serine Avagyan and Elliott Hagedorn for critical reading of the manuscript, Sai Ma for scATAC-Seq analysis advice and critical reading of the manuscript, Maximilian Haeussler and Matthew Speir for help with setting up and hosting the cell browser app and the HCBI, HSCRB FACS core, Office of Animal Resources, and the Bauer Core Facility at Harvard University for technical support. This work was supported by grants from the National Institutes of Health (P01HL131477-04, R01 HL04880, PPG-P015PO1HL32262-32, 5P30 DK49216, 5R01 DK53298, 5U01 HL10001-05, and R24 DK092760) (to LIZ), the Leukemia and Lymphoma Society (Scholar grant 5372–15) (to EMF) and a Boehringer Ingelheim Fonds PhD fellowship (to AS). LIZ is an Investigator of the Howard Hughes Medical Institute.

# Additional information

## Competing interests

David Scadden: is a director and equity holder of Agios Pharmaceuticals, Magenta Therapeutics, Editas Medicines, ClearCreekBio, and Life-VaultBio; a founder of Fate Therapeutics and Magenta Therapeutics; and a consultant to FOG Pharma and VCanBio. Leonard I Zon: is founder and stockholder of Fate, Inc, Scholar Rock, Camp4 therapeutics and a scientific advisor for Stemgent. The other authors declare that no competing interests exist.

## Funding

| Funder | Grant reference number | Author |
|---|---|---|
| National Institutes of Health | R01 HL04880 | Leonard I Zon |
| National Institutes of Health | PPG-P015PO1HL32262-32 | Leonard I Zon |
| National Institutes of Health | 5P30 DK49216 | Leonard I Zon |
| National Institutes of Health | 5R01 DK53298 | Leonard I Zon |
| National Institutes of Health | 5U01 HL10001-05 | Leonard I Zon |
| National Institutes of Health | R24 DK092760 | Leonard I Zon |
| Leukemia and Lymphoma Society | 5372-15 | Eva M Fast |
| Boehringer Ingelheim Fonds | PhD fellowship | Audrey Sporrij |
| National Institutes of Health | P01HL131477-04 | David Scadden Leonard I Zon |

The funders had no role in study design, data collection and interpretation, or the decision to submit the work for publication.

## Author contributions

Eva M Fast, Conceptualization, Formal analysis, Investigation, Software, Visualization, Writing - original draft, Writing – review and editing; Audrey Sporrij, Investigation, Visualization, Writing – review and editing; Margot Manning, Investigation, Writing – review and editing; Edroaldo Lummertz Rocha, Formal analysis; Song Yang, Data curation, Software; Yi Zhou, Project administration, Resources; Jimin Guo, Ninib Baryawno, Methodology; Nikolaos Barkas, Software; David Scadden, Fernando Camargo, Resources, Supervision; Leonard I Zon, Conceptualization, Funding acquisition, Resources, Supervision, Writing – review and editing

## Author ORCIDs

Eva M Fast http://orcid.org/0000-0002-4261-5244
Leonard I Zon http://orcid.org/0000-0003-0860-926X

## Ethics

All animal procedures were approved by the Harvard University Institutional Animal Care and Use Committee (Protocol number 15-03-237).

## Decision letter and Author response

Decision letter https://doi.org/10.7554/eLife.66512.sa1
Author response https://doi.org/10.7554/eLife.66512.sa2

# Additional files

## Supplementary files

- Supplementary file 1. Table of sequencing metrics. Sequencing metric output from cellranger.

- Supplementary file 2. Table with overlap of differentially regulated genes in male and female hematopoietic stem cells (HSCs) and Lin-, c-Kit+, Sca1+ (LSKs). Table summarizing number of differentially regulated genes within male and female HSCs and LSKs. Over-representation analysis odds ratio and p-value were calculated using a Fisher's exact test.

- Supplementary file 3. Table of differential expression result (model-based analysis of single-cell transcriptomics [MAST]) by sex. Each tab contains a treatment vs. control comparison (16,16-dimethyl prostaglandin $E_2$ [dmPGE$_2$], Indo, poly(I:C), granulocyte colony-stimulating factor [G-CSF]). Each cluster was compared to its respective control cluster separated by sex. Log fold change and adjusted p-value from the Hurdle model are listed for genes with p-values < 0.01.

- Supplementary file 4. Table with marker gene enrichments in single-cell RNA sequencing (scRNA-Seq) clusters. Marker gene enrichment was calculated using a Wilcoxon rank-sum test. Score (suffix '_s') indicates the z-score of each gene on which p-value computation is based. Other fields are log fold change = suffix '_l' and false discovery adjusted p-value – suffix '_p'.

- Supplementary file 5. Table with curated pathways used for over-representation analysis. Gene lists curated from literature search and MsigDB.

- Supplementary file 6. Table of pathway enrichment for hematopoietic stem cell (HSC) and Lin-, c-Kit+, Sca1+ (LSK) clusters and treatments. Over-representation analysis for genes induced by external stimulants or enriched in single-cell RNA sequencing (scRNA-Seq) clusters (top 3–5 pathways shown for each database or curated pathway set with adjusted p-value < 0.05). Granulocyte colony-stimulating factor (G-CSF) has an additional tab listing all enrichments (including adjusted p-value > 0.05) of curated pathways.

- Supplementary file 7. Table of differential gene expression result (model-based analysis of single-cell transcriptomics [MAST]). Each tab contains a treatment vs. control comparison (16,16-dimethyl prostaglandin $E_2$ [dmPGE$_2$], Indo, poly(I:C), granulocyte colony-stimulating factor [G-CSF]). Each cluster was compared to its respective control cluster. Log fold change and adjusted p-value from the Hurdle model are listed for genes with p-values < 0.01.

- Supplementary file 8. Table of average expression per cluster of differentially regulated genes in hematopoietic stem cells (HSCs). Count normalized and log transformed UMI counts were averaged across cells in HSC clusters for differentially regulated genes from model-based analysis of single-cell transcriptomics (MAST).

- Supplementary file 9. Table of coocurrence of top 100 genes across Lin-, c-Kit+, Sca1+ (LSK) and hematopoietic stem cell (HSC) treatments and clusters. Comparison of top 100 genes between LSK or HSC treatments and clusters (no duplicates) and comparison of top 100 genes between LSK and HSC clusters (with duplicates) and in HSC clusters Replicate 1 vs. Replicate 2.

- Supplementary file 10. Table of average expression per cluster of differentially regulated genes in Lin-, c-Kit+, Sca1+ (LSKs). Count normalized and log transformed UMI counts were averaged across cells in LSK clusters for differentially regulated genes from model-based analysis of single-cell transcriptomics (MAST).

- Supplementary file 11. Table of ChromVar transcription factor (TF) motif activity score enrichment in Lin-, c-Kit+, Sca1+ (LSK) and hematopoietic stem cell (HSC) single-cell chromatin accessibility sequencing (scATAC) clusters. ChromVar motif activity score enrichment for HSC and LSK scATAC clusters.

- Transparent reporting form

## Data availability

Sequencing data have been deposited in GEO under accession code GSE165844. Processed and integrated single cell data is available here: https://mouse-hsc.cells.ucsc.edu.

The following dataset was generated:

| Author(s) | Year | Dataset title | Dataset URL | Database and Identifier |
|---|---|---|---|---|
| Fast E | 2021 | Niche signals regulate continuous transcriptional states in hematopoietic stem cells | https://www.ncbi.nlm.nih.gov/geo/query/acc.cgi?acc=GSE165844 | NCBI Gene Expression Omnibus, GSE165844 |

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
