## [Decision Letter]

**Decision letter after peer review:**

Thank you for submitting your article "Niche signals regulate continuous transcriptional states in hematopoietic stem cells" for consideration by *eLife*. Your article has been reviewed by 3 peer reviewers, one of whom is a member of our Board of Reviewing Editors, and the evaluation has been overseen by Utpal Banerjee as the Senior Editor. The reviewers have opted to remain anonymous.

Essential revisions:

Expert reviewers have evaluated the manuscript and found the overall work of interest as a resource to the haematopoietic stem cell field, however agreed on a number of issues especially linked to many aspects of its presentation. This includes an unclear rationale of the choice of stressors, the often-misleading reference to niche signals when the HSC niche is not specifically analysed and simple external stimuli are used, and in multiple instances unclear figures. In particular, a unified heatmap added in figure 1 showing differentials across all clusters and perturbations would illustrate what genes identify clusters and what gene expression changes are unique to the perturbations. Next, it will be important to discuss how these gene expression changes either make sense given prior knowledge of the perturbation, or are a surprise. Without this, it is hard to evaluate the potential contribution to the field of the current manuscript.

Specific points:

1) A unified heatmap added in figure 1 showing differentials across all clusters and perturbations would illustrate what genes identify clusters and what gene expression changes are unique to the perturbations. Next, it will be important to discuss how these gene expression changes either make sense given prior knowledge of the perturbation, or are a surprise. Without this, it is hard to evaluate the potential contribution to the field of the current manuscript.

2) Please revise the use of the term 'niche' throughout the manuscript as it is often misleading. The HSC niche is not analysed, and the signals used are stressors administered pharmacologically for as short as two hours. Much better clarity is needed when referring to niche-mediated vs. cell-autonomous responses.

3) Multiple figure panels are unclear – see specific comments from reviewers.

4) The rationale for the choice of stressors is unclear. Please explain it in the introduction.

*Reviewer #1 (Recommendations for the authors):*

It is at odds with all literature that the LKS SLAM CD135- CD34- HSCs are pretty much all in G1, especially when for example 15% of MPP3/4 are in G0. Is this confirmed by the RNAseq data? How could this unusual cell cycle profile be explained?

Supp figure 1. How many mice were analyzed? Error bars are missing in F and G.

The repeated use of the term 'niche stimulation' or 'niche stimulants' is misleading because all factors, even though some can be produced by niche cells and pIpC leads to upregulation of IFNalpha which is known to act both on HSCs and their niche, were administered to the mice. The terms 'stimuli', or 'extrinsic stimuli' seem more appropriate.

Figure 1A. What cells exactly went into the analysis? Is it bulk LKS or bulk LKS plus a number of HSCs to increase their representation? Is Figure 1B showing LKS cells or HSCs? One would think LKS cells, but later the text (lines 124-129) talks about 'HSCs'. The same issue is again in Figure 2E where the figure legend states LKS but main text states HSCs. Do MPPs not show an interferon response following pIpC administration? What does the color coding indicates?

Figure 1B. As the authors point out, the three main clusters appear to be quite unusual, and indeed a continuum of states may be a better explanation for the differences observed. Would it make sense to further develop the visualization of the data to reflect this? For example, cells could be differentially colored based on quiescence/activation/metabolism scores, resulting in a more nuanced picture.

Supp Figure 2E. The gene categories of the quiescent and Acute activation clusters seem highly overlapping. What makes them different?

Line 163 – 163 contradicts the previous section which was largely presented as an analysis of LSK cells, not of HSCs.

Figure 2. It would be helpful to indicate where in the overall plot each control/stimulated sample falls, as done in Figure 1E. One would expect major shifts to become visible between for example control and pIpC treated samples.

Figure 3A. It would be useful to read the authors’ interpretation of the data. Would it be correct to think that G-CSF might have a stronger effect on MPPs, and indo on HSCs?

Figure 3F. Why are there only four LSK clusters, when there were 8 in figure 2A?

Figure 5. The ATACseq data need to be reorganised to make it clearer that differences between HSC 0/1 are from chromatin accessibility of genes downstream of the genes directly affected by the stimuli considered. The same 5 motifs analyzed for HSCs (CRE, STAT, NF-κB, AP-1, ISRE) should be shown for LKS cells too. This is done for some motifs in Figure 5 G-J and is the most helpful comparison.

I wonder if a bioinformatics journal may be a more suitable vehicle for publication of the dataset.

*Reviewer #2 (Recommendations for the authors):*

Specific comments:

1. The results indicate that scRNAseq “in the case of PGE, allowed for a novel transcriptional state to surface”. How is this more novel or distinct than the interferon state that is described?

2. In 2A why rename the “quiescent” cluster as “progenitor” – it is rather misleading because you are taking a gene set enriched for quiescence and calling it progenitor rather than just describing the phenotype. This name is also misleading because it implies that the cells in the other states are not progenitors, which they are.

3. G-CSF exerts well-known effects on HSCs with induction of cell cycle (Schuettpelz 2014) – why is this not evident transcriptionally in the data? This should at minimum be addressed in the discussion.

4. The MPP3 population is supposedly more myeloid biased. It is surprising that this is hardly noticeable in the MPP3/MPP4 population (Figure 2F) – another point that should be addressed in the discussion.

5. Figure 3G and 3H are really difficult to understand. Is the interpretation that cell cycle genes suppressed by PGE are in the HSC and not defined MPP subset of the LSK? What are you calling “within cluster” and what is “between cluster”? Is 3H only showing the cell cycle gene set? Further description and clarification are needed.

6. It is interesting that the interferon cluster can be distinguished from the TLR cluster, but what is more interesting is to understand whether some of the interferon signature genes induced in the other clusters e.g. quiescent, activated, metabolism are due to indirect signaling by IFN to these cells as opposed to direct TLR3 engagement by PIPC. This should be included in the discussion.

*Reviewer #3 (Recommendations for the authors):*

Understanding the molecular determinants controlling hematopoietic stem cell (HSC) biology is critical for myriad clinically-relevant interventions; however, because HSC are rare, this information is limited. Here the authors exploit their considerable facility with HSC isolation and apply single-cell genomics to provide a profile of both normal HSC transcriptional clusters and HSC relevant perturbations (di-methyl-PGE2 vs. the Cox1/2 inhibitor indomethacin, and G-CSF stimulating mobilization, or the TLR3 ligand poly(I:C)) and identify potential underlying regulatory transcription factors based on in silico analyses. They note that they can understand the perturbations as shifts in cells within the unperturbed clusters (with modest gene expression changes in each cluster). There are some aspects of the work that could be changed to improve impact and to clarify the take-home message.

The manuscript leaves the reader with the expectation that the work will biologically dissect the normal and perturbed cluster/populations. This is probably because the authors do not adequately clarify the biological impact of the manipulations, the depth of the published record on them, and then convey the expected versus observed transcriptional changes based on that prior published record. In addition, the transcriptional changes in each cluster within the heatmaps relegated to supplementary data probably provide the essential information, but they fail to represent the data across all clusters with all differentially expressed genes to demonstrate common or distinct gene expression changes. This would best be consolidated to a heatmap of differentials instead of the current method of clustering the actual expression metric. To be clear, it would significantly improve the work to show all differentially expressed genes in each HSC cluster across all perturbed clusters in a single heatmap. A viewer other than a genome browser session (which is not easily maintained) would be an essential improvement.

The central claim is that “niche signals regulate continuous transcriptional states in hematopoietic stem cells”. As an experimental paradigm, the authors inject mice with different molecules and then purify HSC two hours later to examine changes in gene expression. This experimental paradigm does not represent specific perturbations of niche signaling.

As shown by the Morrison, Frenette, and Link laboratories, the same cytokine has different effects on HSC behavior depending on its cellular source. For example, CXCL12 from osteoblasts does not regulate HSC but CXCL12 produced by perivascular cells is essential for HSC maintenance (Nature. 2013 Mar 14; 495(7440): 231-235). It is unclear to this reviewer how super physiological doses of G-CSF, pIpC -which is an artificial analog of dsRNA and never found in normal mice-, or stabilized PGE2 represent physiological perturbations of niche signals. Moreover, the molecules used do not only target HSC but also affect production of cytokines by niche cells. For example, G-CSF results in CXCL12 downregulation in the niche (Leukemia. 2011 Feb;25(2):211-7) whereas pIpC displaces HSC from bone marrow niches (Cell Rep. 2020 Dec 22;33(12):108530). If the authors wish to claim that niche signals regulate transitions between transcriptional states in HSC then they should repeat their analyses in mouse chimeras containing both WT and receptor KO HSC (e.g. WT and IFNγ-/- mice). These will control for cell autonomous vs non-cell autonomous effects before and after G-CSF, pIpC and dmPGE2 injection.

Although HSC clusters were not distinguishable by unique markers, the authors defend a protocol to select optimal parameters that enrich for cell-population markers (non-exclusive) in the Leiden scanpy clusters, reasonably arguing that these represent continuous transitions (though they should acknowledge the limitation of the data sparsity, which might artificially enforce continuity). These data were curated based on prior nomenclature and marker genes. The code, single-cell data and underlying cluster associations are provided in a clear and transparently manner. The authors used standard bioinformatics tools (scanpy, signac, MAST, DPT, chromVar) and web-portal applications (cell browser) to perform their analyses. However, it is worth noting that the authors apply some customized combinations of existing pipelines to address problems. Careful consideration should be paid to when such integrative analyses have not been previously vetted, especially when existing published/benchmarked toolkits have been developed to specifically solve these challenges. In this case, for example, scanpy MNN was developed to correct for batch effects in scanpy clustering and the software cellHarmony for comparing perturbation datasets, finding differential expression patterns among matched single-cell clusters and pathway enrichment for those patterns. However, the custom workflows applied appear to be relatively straight forward and likely to produce consistent results.

Treatment with dmPGE2 gives rise to a novel cluster referred to as 'Acute activation' in the HSC. To qualify this as a derivative to the IER activation cluster (and similarly for the analogous Lsk cluster), the authors should perform reference-based classification to their unperturbed clusters. This is a more quantifiable method to assess the presence or absence of cell populations when a baseline is defined, assuming the perturbations do not impact core cell identity gene expression programs.

The authors note that LSK 'Progenitor' and 'Primed' populations were most like 'Quiescent' and 'Activated' in HSC, but "similar" is not qualified analytically (markers, reference-based classification, gene-set enrichment?).

Is HSC Replicate1 the only sample that was analyzed using 10X Genomics, 3' V2 and are all others were V3? Is this the main reason that combat was needed?

Was MAST performed comparing 10x version 2 and 10x version 3 captures? This would not be recommended even with combat adjustment, especially since the authors have 10x version 3 data for all captures but not 10x version 2. If the authors are able to replicate their findings with the just the 10x version 3, then the results could be considered higher confidence.

The authors mention CITE-Seq which implies the use of ADT conjugated antibodies. The reviewer can only find HTO conjugated antibodies to separate rare MPP populations within the same capture (a cost saving method). Moreover, the authors make note that the hashing worked in a prime figure panel (clearly this is a supplementary figure panel). The authors run the risk of conflating cell hashing with informative cell surface antigen detection to the reader. Hence, the term hashing should be used instead of CITE-Seq.

The analysis of the scATAC seq is extremely limited, and hence the value of the inferred regulators needs to be moderated in the text.

[Editors' note: further revisions were suggested prior to acceptance, as described below.]

Thank you for resubmitting your work entitled "External signals regulate continuous transcriptional states in hematopoietic stem cells" for further consideration by *eLife*. Your revised article has been evaluated by Utpal Banerjee (Senior Editor) and a Reviewing Editor.

The manuscript has been improved but there are some remaining issues that need to be addressed, as outlined below:

The reviewers have agreed on a few sentences that would benefit from further clarification or some edits. Specifically:

1) The figure labels seem to be off: Figure 2C does not seem to correspond to the description confirming surface phenotypes with CITE-seq? Description of LSK cells labeled Figure 2C, D, etc in the text (page 6) seems to correspond to the current Figure 3.

2) Page 19, Line ~600-there is a discussion of HSC cluster 0 and cluster 1 but these clusters have not been previously introduced. Are they simply based on the ATACseq data, or do they also correspond to some clustering in the RNAseq?

3) Discussion:

"While we could not detect interferon ligands in our scRNA-Seq data of 723 HSCs or MPPs" – what is meant by "interferon ligands"? Interferons?

4) 2-4 orders of magnitude higher than what an animal would ever encounter during… probably one should say "2-4 orders of magnitude higher than what an animal would typically encounter…." "ever" is a strong word.

---

## [Author Response]

Essential revisions:Expert reviewers have evaluated the manuscript and found the overall work of interest as a resource to the haematopoietic stem cell field, however agreed on a number of issues especially linked to many aspects of its presentation. This includes an unclear rationale of the choice of stressors, the often-misleading reference to niche signals when the HSC niche is not specifically analysed and simple external stimuli are used, and in multiple instances unclear figures. In particular, a unified heatmap added in figure 1 showing differentials across all clusters and perturbations would illustrate what genes identify clusters and what gene expression changes are unique to the perturbations. Next, it will be important to discuss how these gene expression changes either make sense given prior knowledge of the perturbation, or are a surprise. Without this, it is hard to evaluate the potential contribution to the field of the current manuscript.

We thank the reviewers for the thoughtful and detailed feedback on our manuscript. We also thank the reviewers and *eLife* staff for the flexibility around timing of resubmission during the COVID-19 pandemic. In accordance with the reviewers’ suggestions, we have substantially reorganized our manuscript and addressed all reviewer comments. Please find our point by point responses below. We hope that with these revisions the reviewers will find our work appropriate for publication in *eLife*.

Specific points:1) A unified heatmap added in figure 1 showing differentials across all clusters and perturbations would illustrate what genes identify clusters and what gene expression changes are unique to the perturbations. Next, it will be important to discuss how these gene expression changes either make sense given prior knowledge of the perturbation, or are a surprise. Without this, it is hard to evaluate the potential contribution to the field of the current manuscript.

As requested, a unified heatmap showing differentials of both clusters and treatments is now added to Figure 1 for HSCs (Figure 1D) and a similar heatmap has been added to Figure 3 (Figure 3E) for LSK cells. We choose to display 4 representative genes for each cluster or treatment in an effort to increase readability of the figures. In addition, we have added a larger heatmap to Figure 1 —figure supplement 2I. To further identify unique or common gene signatures, we have included pairwise comparisons of the top 100 genes enriched in each cluster and following treatment for HSCs (Figure 1H) and LSKs (Figure 3F). All the underlying data is available via easily searchable source files (Supplementary file 4, 7-10). To compare our results to prior studies, we have expanded our over-representation analysis to more pathways (GO:BP and REACTOME) and have added 49 manually curated gene sets from literature search and the Molecular Signatures Database (MSigDB) (Supplementary file 5 and 6). Based on these new and refined analyses, we describe the following main novel insights of HSC response to external stimulants.

– Our analysis showed that part of the gene signatures induced by dmPGE_2_ or poly(I:C) matched previous reports. Selected G-CSF perturbed genes (*Spi1, Myb, Nurp, ckit, Cd9*) were consistent with previous knowledge. However, our G-CSF induced gene signature did not show enrichment with previous reports of G-CSF treated HSCs (both scRNA-Seq and bulk RNA-Seq), likely because of a difference in timing of treatment (2h vs 36h). Overall, our stimulant induced differentially expressed genes show some concordance to previous publications but also reveal new genes that may be specific for the short (2 hour) treatment.

– The new analysis comparing genes enriched in clusters and treatments highlighted a difference in transcriptional state and response between LSKs and HSCs. In contrast to LSKs, HSC baseline transcriptional heterogeneity is increasingly determined by different signaling gradients. This was evident by the large overlap of baseline cluster identity genes with stimulant induced genes (Figure 1H vs Figure 3F). Comparison of the total number of induced genes between HSCs and LSKs indicated further specificity of HSC response to external stimuli. We observed differences in preferred cell type for each stimulant (poly(I:C) – HSCs, G-CSF – LSKs) and magnitude of overall response (indomethacin – small changes).

– We discovered that HSC response to treatment with external stimulants is heterogeneous even within a given treatment – similar to our observation of fluent transcriptional states at baseline. The unified heatmap in Figure 1D for example illustrates that HSC clusters respond differently to poly(I:C). Even though all poly(I:C) treated HSC clusters show some induction of poly(I:C) target genes, response is most strong in the Interferon cluster. Our scATAC-seq analysis provides preliminary evidence that cell intrinsic chromatin heterogeneity at baseline may regulate responsiveness to external signals.

– Our data provides a rich resource for a number of additional analyses and replication studies that would have exceeded the space allocation of the current manuscript.

Examples of these analysis would be:

– Further identification of unique responses within clusters for each treatment (Analysis of cell cycle genes in dmPGE_2_ treated LSK cells of dmPGE_2_ which was removed from the updated manuscript in favor of streamlining discussion of HSC relevant results).

– Evaluation of genes that follow opposite directionality between HSCs and LSKs (IEGs in dmPGE_2_).

– Comparison of commonly and uniquely regulated genes between stimulants.

– Expansion of trajectory analyses within other treatments (Metabolism cluster – G-CSF).

– Effects of external stimulation on myeloid fate (myeloid cluster cells are predominantly from non-control cells).

– In depth analysis of sexual dimorphism.

2) Please revise the use of the term 'niche' throughout the manuscript as it is often misleading. The HSC niche is not analysed, and the signals used are stressors administered pharmacologically for as short as two hours. Much better clarity is needed when referring to niche-mediated vs. cell-autonomous responses.

We agree with the reviewers about the ambiguity of our previous terminology and have replaced ‘niche stimulation’ with ‘external stimulation’ or similar terminology wherever applicable throughout the manuscript.

3) Multiple figure panels are unclear – see specific comments from reviewers.

We thank the reviewers for the multiple specific comments related to the presentation of our data. These questions helped us streamline and clarify the main conclusions of our work. We have taken the following specific actions and substantially reorganized our manuscript.

– We have switched the order of the figures so that Figure 1 and 2 only discuss HSCs and Figure 3 and 4 contain the comparative analysis of LSKs versus HSCs. To better communicate this layout to the reader we have updated the experimental schematic in Figure 1A indicating which Figure contain which cell type.

– We have improved terminology to increase clarity. For example, to further clarify which clusters belong to HSCs and which clusters belong to LSKs we have added a cell type specific prefix ‘LSK-’ to all LSK clusters. We have renamed the MPP (LSK, CD48-, CD150-) population into MPP0 to avoid confusion between all MPPs and this particular subset.

– We have reorganized Figure 3 (LSKs) so that the layout is similar to Figure 1 (HSCs). That way the reader should more easily be able to compare and contrast stimuli responses in HSCs versus LSKs.

– We have substantially revised the text to improve readability.

– Following two reviewers’ comments we have simplified the ATAC-seq section and moderated our conclusions.

4) The rationale for the choice of stressors is unclear. Please explain it in the introduction.

In the updated manuscript we have included the following sentence outlining the rationale for our stimulants in the main text:

‘To encompass a wide variety of different transcriptional responses, we evaluated three different signaling pathways: an inflammatory pathway through stimulation or inhibition of prostaglandins by dmPGE_2_ and indomethacin, a host-defense immune signaling pathway mediated by activating of TLR and interferon signaling with poly(I:C), and a cellular mobilization pathway stimulated by the growth factor G-CSF.’

We have further added a paragraph to the discussion explaining the rationale for using pharmacological perturbations of the niche and potential drawbacks of our study.

‘We evaluated the effect of three complementary signaling pathways (G-CSF, Prostaglandin and Interferon) on the transcriptional state of HSCs. Pharmacological perturbation of these signaling pathways allowed to tightly control critical experimental parameters (e.g. genetic background of mice, timing of sample processing) that mitigated potential confounders of the downstream analysis. With the exception of indomethacin, we chose a short treatment window of two hours to increase the likelihood of studying direct downstream effects of stimulants on HSCs. Analysis of DEGs within clusters indicated interferon- versus toll-like receptor response genes induced by poly(I:C) treatment. While we could not detect interferon ligands in our scRNA-Seq data of HSCs or MPPs, it is possible that some of the interferon-response genes were induced indirectly by release of IFN ligands from the niche. An interferon inducer similar to poly(I:C) has been previously shown to increased IFN-α protein levels in the serum as early as 2 hours post in vivo injection (Linehan et al., 2018). Future work using genetic models is needed to further dissect indirect versus direct effects of external stimulants on HSCs.‘

Reviewer #1 (Recommendations for the authors):It is at odds with all literature that the LKS SLAM CD135- CD34- HSCs are pretty much all in G1, especially when for example 15% of MPP3/4 are in G0. Is this confirmed by the RNAseq data? How could this unusual cell cycle profile be explained?

We thank the reviewer for the careful and critical evaluation of all our data. Although we were not able to determine the ultimate cause of the discrepancy of our cell-cycle results and previously published work, we suspect a technical artifact during preparation of our samples (ki67 antibody concentration). We therefore decided to remove the cell cycle experiment from the updated manuscript. We added further evidence showing pathway enrichment of Quiescent HSCs (Supplementary file 6). Together with our LDTA data we believe that we provide sufficient data demonstrating that our purified HSCs adequately match previously reported ones.

Supp figure 1. How many mice were analyzed? Error bars are missing in F and G.

We have added the number of mice to the Figure legend of Figure1 —figure supplement 1 (previous Supp figure 1). Since each condition (stacked barplot) represents a pooled sort of 5 mice we were not able to calculate variability of cell proportions between individual mice. We deliberately opted against keeping individual mice separate during sample preparation because it would have significantly increased the processing time (because of increased tube handling and limitations of our lineage depletion equipment) and thus could have compromised cell viability for the single cell analysis.

The repeated use of the term 'niche stimulation' or 'niche stimulants' is misleading because all factors, even though some can be produced by niche cells and pIpC leads to upregulation of IFNalpha which is known to act both on HSCs and their niche, were administered to the mice. The terms 'stimuli', or 'extrinsic stimuli' seem more appropriate.

We thank the reviewer for this critical suggestion. The terminology of ‘niche stimulants’ was replaced with ‘extrinsic stimulants’ wherever appropriate.

Figure 1A. What cells exactly went into the analysis? Is it bulk LKS or bulk LKS plus a number of HSCs to increase their representation? Is Figure 1B showing LKS cells or HSCs? One would think LKS cells, but later the text (lines 124-129) talks about 'HSCs'. The same issue is again in Figure 2E where the figure legend states LKS but main text states HSCs. Do MPPs not show an interferon response following pIpC administration? What does the color coding indicates?

For our HSC scRNA-Seq analysis, we use purified HSC cells. For the LSK scRNA-Seq analysis, we pooled MPPs and HSCs in the same ratios that we would observe during FACS. The proportion of HSCs in LSK analyses are therefore very small (~2% of total cells).

In order to more clearly indicate which analyses include HSCs and which include LSK progenitors we made the following changes:

– We created an updated flowchart in Figure 1A that specifically highlights which figure contains which cell type.

– We have reorganized the figures so that Figure 1 and 2 exclusively contain data on HSCs and Figure 3 and 4 is the comparative analysis of HSCs versus LSKs.

– We changed the cluster nomenclature in LSKs by adding an ‘LSK-’ prefix to more clearly identify similar clusters arising in both HSCs and LSKs.

Figure 1B. As the authors point out, the three main clusters appear to be quite unusual, and indeed a continuum of states may be a better explanation for the differences observed. Would it make sense to further develop the visualization of the data to reflect this? For example, cells could be differentially colored based on quiescence/activation/metabolism scores, resulting in a more nuanced picture.

We thank the reviewer for this very insightful suggestion. We have calculated scores for each HSC cluster by combining all genes enriched in clusters. Specifically, our score represents the average expression of a set of marker genes subtracted by the average expression of a reference set of genes (described in detail in the methods). We have included UMAP visualizations of these scores in Figure 1C to better illustrate the transcriptional continuum of HSCs.

Supp Figure 2E. The gene categories of the quiescent and Acute activation clusters seem highly overlapping. What makes them different?

We thank the reviewer for the careful evaluation of our supplemental figures. To further analyze gene sets enriched in our clusters and clear up any ambiguities we have modified Figure 2E and instead produced supplementary file 6 that contains a more comprehensive pathway enrichment analysis. In addition to REACTOME and GO:BP pathways we also evaluate enrichment of gene sets specific to HSCs which were previously linked to a cycling status (Cabezas-Wallscheid et al., 2017) or functional potential (Rodriguez-Fraticelli et al., 2020). Using this more refined methodology we were able to verify that the names we assigned to our clusters (e.g. ‘Quiescent’, ‘Metabolism’) do indeed match previously reported HSC signatures and unambiguously identify different groups. Our updated pathway and gene set enrichment procedure is explained in detail in the Methods section.

Line 163 – 163 contradicts the previous section which was largely presented as an analysis of LSK cells, not of HSCs.

We have updated the figures, section headings and cluster names to clarify which cell type is being discussed in which section.

Figure 2. It would be helpful to indicate where in the overall plot each control/stimulated sample falls, as done in Figure 1E. One would expect major shifts to become visible between for example control and pIpC treated samples.

In previous Figure 2 (new Figure 3G) we have included a density plot of treated LSKs that is similar to the density plot of treated HSCs in Figure 1E. We do indeed see a cellular shift of poly(I:C) treated cells compared to control.

Figure 3A. It would be useful to read the authors' interpretation of the data. Would it be correct to think that G-CSF might have a stronger effect on MPPs, and indo on HSCs?

Figure 3A (new Figure 4A-D) illustrates the proportion of differentially expressed genes (at different fold change cutoffs) for each individual treatment (G-CSF, indo, poly (I:C) and dmPGE_2_) in HSCs and LSKs. Furthermore, we have classified genes into being regulated exclusively in HSCs and LSKs or being induced/downregulated in both cell types. Relatively speaking there is a higher proportion of unique genes changed by indo in HSCs (1.2 fold cut-off: 100% HSCs) and conversely in LSKs by G-CSF (1.2 fold cut-off: 68% LSKs). However, when comparing the absolute number of genes perturbed G-CSF and indo, G-CSF still changes more genes in HSCs than indo (51 vs 21). Indomethacin overall induces very minor changes in gene expression (Figure 4 and Figure 4 Source Table 1).

Figure 3F. Why are there only four LSK clusters, when there were 8 in figure 2A?

When aggregating gene expression for particular clusters/conditions we did not include clusters that have less than 20 cells because of the noisy nature of scRNA-Seq measurements. LSK control cells were composed of the four main clusters Primitive, Primed, Metabolism and Cell-cycle. We were only able to detect 75 myeloid cells overall (including all external pertubations). The control condition only contained two myeloid cells. We therefore left myeloid cells out from Figure 4F. The three clusters Interferon, Interferon Cell-cycle and Acute-Activation did only exist in the treated conditions and therefore these clusters were not considered for receptor expression at baseline. We have included language in the legend that explains that only clusters with more than 20 cells are shown.

Figure 5. The ATACseq data need to be reorganised to make it clearer that differences between HSC 0/1 are from chromatin accessibility of genes downstream of the genes directly affected by the stimuli considered. The same 5 motifs analyzed for HSCs (CRE, STAT, NF-κB, AP-1, ISRE) should be shown for LKS cells too. This is done for some motifs in Figure 5 G-J and is the most helpful comparison.

According to the reviewers’ suggestion we have reorganized the scATAC-Seq results. We have simplified Figure 5 so that we now show violin plots for differentially accessible motifs in HSCs (Figure 5D) and the same motifs in LSKs (Figure 5E) directly below. For clarity and to streamline the results, we have removed the data on CTCF, YY1, NRF1 motifs.

I wonder if a bioinformatics journal may be a more suitable vehicle for publication of the dataset.

We respectfully disagree with the reviewer that a bioinformatics journal is more suitable for our work. We did not develop any new computational methods and used fairly standard analysis tools. We expect that our resource will be primarily used by researchers interested in stem cell regeneration and/or hematopoiesis. We selected *eLife* is an appropriate journal for our work because of the broad readership and its open-source policy.

Reviewer #2 (Recommendations for the authors):Specific comments:1. The results indicate that scRNAseq "in the case of PGE, allowed for a novel transcriptional state to surface". How is this more novel or distinct than the interferon state that is described?

We have observed a small but reproducible population of HSCs in the interferon cluster even in control mice in the absence of external stimulation (Figure 1 —figure supplement 2A-D). Upon poly(I:C) treatment the proportion of cells within the ‘Interferon’ cluster increases from 1% to 42%. In contrast no ‘Acute activation’ cluster exists at baseline. We therefore concluded that dmPGE_2_ treatment led to the formation of a novel transcriptional state.

2. In 2A why rename the "quiescent" cluster as "progenitor" – it is rather misleading because you are taking a gene set enriched for quiescence and calling it progenitor rather than just describing the phenotype. This name is also misleading because it implies that the cells in the other states are not progenitors, which they are.

We compared genes enriched in clusters between HSCs and LSKs to detect some common signatures that would allow us to better understand the identity of these clusters. For example, 86 genes out of the top 100 genes were shared between the ‘Interferon’ clusters in HSCs and LSKs (Figure 3A). For other clusters we observed a lower similarity and often not a one-to-one match. The LSK Progenitor cluster actually shared genes with both the HSC ‘Activated’ and the HSC ‘Quiescent’ cluster. To avoid suggesting that clusters between HSCs and LSKs are equivalent we added a prefix of ‘LSK-’ to all LSK clusters. Accordingly, the following sentence was added

‘Because the latter analysis only indicated similarity rather than full equivalence of HSC and LSK clusters, and to avoid ambiguity when evaluating HSCs and LSKs, all LSK clusters were denoted with the prefix ‘LSK-’.’

To avoid confusion between the terms LSK progenitors and the LSK-Progenitor cluster we have renamed the cluster into “LSK-Primitive”.

3. G-CSF exerts well-known effects on HSCs with induction of cell cycle (Schuettpelz 2014) – why is this not evident transcriptionally in the data? This should at minimum be addressed in the discussion.

We deliberately chose to assess transcriptional changes very early (2 hours) after G-CSF treatment to observe transcriptional response directly downstream of the stimulant. To address the earlier question about known versus novel transcriptional regulation of the different stimulants (see above) we performed enrichment tests with curated gene sets including the one from Schuettpelz, et al. 2014. We did not see a statistically significant overlap of our G-CSF induced gene expression signature and the one from Schuettpelz (Supplementary file 5 and 6). A likely explanation is that the G-CSF gene expression program significantly changes between 2h and 36h post treatment. Our data shows that G-CSF activates a metabolism program that likely results in increased cycling activity. Indeed, in our heatmap of all G-CSF differentially expressed genes (Figure 1 —figure supplement 4B) the ‘Metabolism’ cluster of G-CSF treated cells groups together with the Cell-cycle cluster.

We have updated the following text in the results:

‘However, our G-CSF induced gene set did not show any significant enrichment (Supplementary file 5 and 6) with various previously reported G-CSF signatures (Schuettpelz et al., 2014, Pedersen et al., 2016, Giladi et al., 2018, Mervosh et al., 2018) likely due to different timing of G-CSF treatment.’

and

‘Hierarchical clustering suggested that G-CSF treatment drove the expression profile of the HSC ‘Metabolism’ cluster closer towards the ‘Cell cycle’ state (Figure 1—figure supplement 4B).’

4. The MPP3 population is supposedly more myeloid biased. It is surprising that this is hardly noticeable in the MPP3/MPP4 population (Figure 2F) – another point that should be addressed in the discussion.

While MPP2s contain the largest proportion of myeloid cells the myeloid cluster itself is made up of 71% of MPP3/4 (Figure 3 —figure supplement 1G, Figure 3 Source Table 1). This is because overall MPP3/4 make up a much larger part of the MPP population than MPP2s (Figure 1 –supplement 1F) We recognize that we failed to sufficiently explain this observation in the original version of the manuscript and have now added the following sentence to the main text

‘Consistent with previous reports (Pietras et al., 2015), our data indicated that the ‘LSK-Myeloid’ cluster was composed of MPP2s and MPP3/4 cells but no HSCs, MPP0s or MPP1s (Figure 3—figure supplement 1G).’

and discussion:

‘For example, even though both MPP2 and MPP3 cells have been previously described as myeloid biased (Pietras et al., 2015) our analysis allowed to determine the proportion of putative myeloid cells within MPP2 and MPP3/4 cells as well as the relative MPP2 and MPP3/4 composition of myeloid cells.’

5. Figure 3G and 3H are really difficult to understand. Is the interpretation that cell cycle genes suppressed by PGE are in the HSC and not defined MPP subset of the LSK? What are you calling "within cluster" and what is "between cluster"? Is 3H only showing the cell cycle gene set? Further description and clarification are needed.

We have chosen to take this analysis out of our update manuscript. In the new version we focused on better explaining existing data and in particular highlighting HSC specific transcriptional responses to stimuli. Since dmPGE_2_ effect on cell cycle genes is an observation only seen in LSKs we have chosen to remove these results from the final manuscript.

6. It is interesting that the interferon cluster can be distinguished from the TLR cluster, but what is more interesting is to understand whether some of the interferon signature genes induced in the other clusters e.g. quiescent, activated, metabolism are due to indirect signaling by IFN to these cells as opposed to direct TLR3 engagement by PIPC. This should be included in the discussion.

According to the reviewers’ suggestion we have included a new paragraph in the discussion about timing of perturbation and evaluating indirect versus direct signaling with IFN and TLR3 as an example. Further rationale for our pharmacological perturbation of niche signaling is given.

‘We evaluated the effect of three complementary signaling pathways (G-CSF, Prostaglandin and Interferon) on the transcriptional state of HSCs. Pharmacological perturbation of these signaling pathways allowed to tightly control critical experimental parameters (e.g. genetic background of mice, timing of sample processing) that mitigated potential confounders of the downstream analysis. With the exception of indomethacin, we chose a short treatment window of two hours to increase the likelihood of studying direct downstream effects of stimulants on HSCs. Analysis of DEGs within clusters indicated interferon- versus toll-like receptor response genes induced by poly(I:C) treatment. While we could not detect interferon ligands in our scRNA-Seq data of HSCs or MPPs, it is possible that some of the interferon-response genes were induced indirectly by release of IFN ligands from the niche. An interferon inducer similar to poly(I:C) has been previously shown to increased IFN-α protein levels in the serum as early as 2 hours post in vivo injection (Linehan et al., 2018). Future work using genetic models is needed to further dissect indirect versus direct effects of external stimulants on HSCs.’

Reviewer #3 (Recommendations for the authors):Understanding the molecular determinants controlling hematopoietic stem cell (HSC) biology is critical for myriad clinically-relevant interventions; however, because HSC are rare, this information is limited. Here the authors exploit their considerable facility with HSC isolation and apply single-cell genomics to provide a profile of both normal HSC transcriptional clusters and HSC relevant perturbations (di-methyl-PGE2 vs. the Cox1/2 inhibitor indomethacin, and G-CSF stimulating mobilization, or the TLR3 ligand poly(I:C)) and identify potential underlying regulatory transcription factors based on in silico analyses. They note that they can understand the perturbations as shifts in cells within the unperturbed clusters (with modest gene expression changes in each cluster). There are some aspects of the work that could be changed to improve impact and to clarify the take-home message.The manuscript leaves the reader with the expectation that the work will biologically dissect the normal and perturbed cluster/populations. This is probably because the authors do not adequately clarify the biological impact of the manipulations, the depth of the published record on them, and then convey the expected versus observed transcriptional changes based on that prior published record. In addition, the transcriptional changes in each cluster within the heatmaps relegated to supplementary data probably provide the essential information, but they fail to represent the data across all clusters with all differentially expressed genes to demonstrate common or distinct gene expression changes. This would best be consolidated to a heatmap of differentials instead of the current method of clustering the actual expression metric. To be clear, it would significantly improve the work to show all differentially expressed genes in each HSC cluster across all perturbed clusters in a single heatmap. A viewer other than a genome browser session (which is not easily maintained) would be an essential improvement.The central claim is that "niche signals regulate continuous transcriptional states in hematopoietic stem cells". As an experimental paradigm, the authors inject mice with different molecules and then purify HSC two hours later to examine changes in gene expression. This experimental paradigm does not represent specific perturbations of niche signaling.

We thank the reviewer for the critical and constructive feedback of our manuscript. We further value the assessment that ‘understanding the molecular determinants controlling hematopoietic stem cell (HSC) biology is critical for myriad clinically-relevant interventions’ which was one of the driving forces for us to undertake this investigation. We have substantially reorganized our manuscript and added additional analysis responding to the concerns raised by the reviewer.

Specifically, we have compared our stimulant induced gene signatures to prior publications to provide additional context for our results in light of previous findings. As suggested, we have compiled a unified heatmap (Figure 1D) showing differentially expressed genes between clusters and treatments, which provided additional insights into the crosstalk between cluster defining and treatment induced genes. We have chosen to only display selected genes as opposed to all differentially expressed genes in the main Figure, to increase readability and allow easy referencing from the main text. We have added a heatmap encompassing a larger number of genes to Figure 1 —figure supplement 2I. In addition, we have added visualizations of pairwise comparisons of cluster-defining and stimulant-induced genes (Figure 1H and 3F). Source tables contain the complete set of differentially regulated genes for treatments and clusters.

We have deliberately chosen not to add another interactive visualization application to this manuscript. Currently our data is hosted externally for interactive exploration on the UCSC Cell Browser website (https://cells.ucsc.edu/) which provides a free resource for scientists to make their single-cell datasets available (378 single-cell datasets – July 2021). In addition, we plan to make all source datasets (such as differential expression analyses, cluster enrichments, cluster-treatment overlaps) available in a tabular format that ensures both persistence into the future as well as easy data accessibility for non-computational biologists.

We agree that the original terminology of ‘niche stimulants regulating HSC transcriptional states’ was not fully accurate. We have revised this terminology throughout the manuscript to ‘external stimulation’ or equivalent wording. While our pharmacological perturbations certainly have limitations (discussed in detail below and in the discussion) we do believe that our results provide novel findings about HSC response to external stressors and the relationship to baseline transcriptional heterogeneity. Because of the cost and time required of single cell genomics studies we believe that our work serves as an important starting ground for more fine-tuned investigations of the niche-HSC interaction using genetic models.

As shown by the Morrison, Frenette, and Link laboratories, the same cytokine has different effects on HSC behavior depending on its cellular source. For example, CXCL12 from osteoblasts does not regulate HSC but CXCL12 produced by perivascular cells is essential for HSC maintenance (Nature. 2013 Mar 14; 495(7440): 231-235). It is unclear to this reviewer how super physiological doses of G-CSF, pIpC -which is an artificial analog of dsRNA and never found in normal mice-, or stabilized PGE2 represent physiological perturbations of niche signals. Moreover, the molecules used do not only target HSC but also affect production of cytokines by niche cells. For example, G-CSF results in CXCL12 downregulation in the niche (Leukemia. 2011 Feb;25(2):211-7) whereas pIpC displaces HSC from bone marrow niches (Cell Rep. 2020 Dec 22;33(12):108530). If the authors wish to claim that niche signals regulate transitions between transcriptional states in HSC then they should repeat their analyses in mouse chimeras containing both WT and receptor KO HSC (e.g. WT and IFNγ-/- mice). These will control for cell autonomous vs non-cell autonomous effects before and after G-CSF, pIpC and dmPGE2 injection.

We agree with the author that the original terminology of ‘niche stimulation’ was not fully accurate. Even though we did deliberately choose a short treatment regimen (2 hours) to assess likely direct effects of our perturbations on HSCs we cannot rule out that some of our transcriptional responses are induced indirectly through other cells. While we agree that the reviewer’s proposed experiment with receptor knockouts are best to resolve cell autonomous versus non-cell autonomous effects, we think these investigations are beyond the scope of this manuscript. We do believe there is value in reporting transcriptional changes with our chosen external perturbations since they are also widely used in experimental protocols to induce HSC mobilization (G-CSF) or transgene induction (pol(I:C) – mx-cre). Because reviewer #2 expressed a similar concern about direct vs indirect signaling we have updated the discussion with the following paragraph:

‘We evaluated the effect of three complementary signaling pathways (G-CSF, Prostaglandin and Interferon) on the transcriptional state of HSCs. Pharmacological perturbation of these signaling pathways allowed to tightly control critical experimental parameters (e.g. genetic background of mice, timing of sample processing) that mitigated potential confounders of the downstream analysis. With the exception of indomethacin, we chose a short treatment window of two hours to increase the likelihood of studying direct downstream effects of stimulants on HSCs. Analysis of DEGs within clusters indicated interferon- versus toll-like receptor response genes induced by poly(I:C) treatment. While we could not detect interferon ligands in our scRNA-Seq data of HSCs or MPPs, it is possible that some of the interferon-response genes were induced indirectly by release of IFN ligands from the niche. An interferon inducer similar to poly(I:C) has been previously shown to increased IFN-α protein levels in the serum as early as 2 hours post in vivo injection (Linehan et al., 2018). Future work using genetic models is needed to further dissect indirect versus direct effects of external stimulants on HSCs.‘

We furthermore agree with the author’s assessment of the limitations of using super physiological doses. Including orally dosed indomethacin (through the drinking water) in our original experimental design addressed that same concern. We have included a section in the discussion on the tradeoffs between analyzing physiological perturbations with potentially small effect sizes versus strong external stimulants that lead to more robust experimental results.

‘There is a trade-off between the strength of a perturbation required for experimental robustness versus studying signals that are more physiologically relevant but lead to more subtle changes within and between cells. Here, we evaluated response of HSCs to three different external activators mimicking niche signals that were dosed 2-4 orders of magnitude higher than what an animal would ever encounter during actual injury or infection (Eyles et al., 2008, Porter et al., 2013, Hoggatt et al., 2013, Sheehan et al., 2015). To assess niche-derived signals in a more physiological setting, we administered the Cox1/2 inhibitor indomethacin orally for one week to deplete endogenous prostaglandins. As expected, the changes in gene expression with indomethacin were much weaker than those observed after acute injection with dmPGE_2_, G-CSF, and poly(I:C). ScRNA-Seq analysis offers unique tools to evaluate gene expression changes in response to weak perturbations. Pseudotime analysis showed that depletion of endogenous prostaglandins using indomethacin led to a small but significant shift in the transcriptional state of HSCs.’

Although HSC clusters were not distinguishable by unique markers, the authors defend a protocol to select optimal parameters that enrich for cell-population markers (non-exclusive) in the Leiden scanpy clusters, reasonably arguing that these represent continuous transitions (though they should acknowledge the limitation of the data sparsity, which might artificially enforce continuity).

Based on this reviewer’s and Reviewer #1s comments we have expanded and clarified our observation that HSC clusters represent marker gene enrichments but not exclusive expression (Figure 1C). We have further clarified that using Leiden clustering is more of an analytic tool as opposed to an exclusive classification of cell state in the main text.

‘Transcriptional scores visualized that these HSC states were not exclusive, and that HSC transcriptional state could be rather described by a combination of continuous gradients of marker genes. Therefore, subsequent analyses via discrete clusters provided an analytical tool to compare changes in transcriptional state as opposed to an exclusive assignment of cell identities.’

According to this reviewer’s suggestion we have added additional language to the discussion acknowledging sampling limitations in scRNASeq.

‘While we cannot entirely rule out that the continuous cell states arose from the noisy nature of scRNA-Seq sampling, this is unlikely given our observation that genes that vary along the same transcriptional gradients are also functionally correlated (e.g. IEGs).’

These data were curated based on prior nomenclature and marker genes. The code, single-cell data and underlying cluster associations are provided in a clear and transparently manner. The authors used standard bioinformatics tools (scanpy, signac, MAST, DPT, chromVar) and web-portal applications (cell browser) to perform their analyses.

We thank the reviewer for these comments. We strive not only for rigorous analysis of our datasets but also to transparently document each step of the analysis. We furthermore made all of our data and analyses (including intermediate steps) available for other researchers to reproduce (Github and Dockerhub).

However, it is worth noting that the authors apply some customized combinations of existing pipelines to address problems. Careful consideration should be paid to when such integrative analyses have not been previously vetted, especially when existing published/benchmarked toolkits have been developed to specifically solve these challenges. In this case, for example, scanpy MNN was developed to correct for batch effects in scanpy clustering and the software cellHarmony for comparing perturbation datasets, finding differential expression patterns among matched single-cell clusters and pathway enrichment for those patterns.

For batch correction, clustering and differential expression analysis we followed a best practice example (M.D. Luecken, F.J. Theis, Molecular Systems Biology 15(6) (2019): e8746) that was most current when we initiated data analysis for this project. As part of the revisions, we included additional rationale (Methods) and analysis documentation (Github) for choosing ComBat for batch correction over other methods such as Harmony and Scanoramy. We did attempt to install the cellHarmony tool but unfortunately failed to integrate the python 2.7 framework, that cellHarmony is built on, into our workflow (Docker containers).

However, the custom workflows applied appear to be relatively straight forward and likely to produce consistent results.

We thank the reviewer for this assessment.

Treatment with dmPGE2 gives rise to a novel cluster referred to as 'Acute activation' in the HSC. To qualify this as a derivative to the IER activation cluster (and similarly for the analogous Lsk cluster), the authors should perform reference-based classification to their unperturbed clusters. This is a more quantifiable method to assess the presence or absence of cell populations when a baseline is defined, assuming the perturbations do not impact core cell identity gene expression programs.The authors note that LSK 'Progenitor' and 'Primed' populations were most like 'Quiescent' and 'Activated' in HSC, but "similar" is not qualified analytically (markers, reference-based classification, gene-set enrichment?).

We agree with the reviewer and have provided additional analytical qualification for our comparisons of cluster similarity. Specifically we have included the following two approaches in the updated manuscript:

1. As in the original version of the manuscript we have compared the top 100 genes enriched in HSC and LSK clusters. In the updated manuscript we have included this comparison in the main figures (Figure 3A) to increase visibility and performed formal enrichment tests of the pairwise cluster comparison (hypergeometric test). We have deliberately chosen a fixed number of top genes (100) for each cluster as opposed to a fold-change cut-off to allow comparisons of the number of common genes between individual pairwise overlaps.

2. We have calculated ‘scores’ from top genes enriched in each cluster (see Methods) an analysis that was also used to demonstrate gradual changes in transcriptional identity (Figure 1C). To compare similarity of clusters we compared mean scores between all clusters. For example, we calculated the mean ‘Activated’ score for HSCs in all clusters and found that the ‘Acute-Activation’ cluster had the highest mean score besides the ‘Activated’ cluster itself (Figure 1 —figure supplement 2G). Since any gene set can be used to calculate a score, we have used genes enriched in the HSC ‘Quiescent’ cluster to calculate scores in the LSK population. This analysis was used to qualify the statement that the ‘LSK-Primed’ and ‘LSK-Primitive’ (former ‘LSK-Progenitor’) cluster are most similar to the HSC ‘Quiescent cluster (Figure 3 —figure supplement 1H). We have also calculated an HSC ‘Activated’ score in the LSK cluster. Actually the ‘Acute activation’ and ‘Progenitor’ cluster had the highest HSC ‘Activated’ score and not the ‘Primed’ cluster (results available on GitHub). We therefore removed similarity of ‘Primed’ ‘Progenitor’ with the HSC ‘Activated’ cluster from the main text.

Is HSC Replicate1 the only sample that was analyzed using 10X Genomics, 3' V2 and are all others were V3? Is this the main reason that combat was needed?

It is true that HSC Replicate1 was the only sample analyzed with 10x Genomics 3’ V2 and all other samples are using V3. HSC Replicate1 was not combined with any V3 dataset and only analyzed separately, shown in the following Figures (Figure 1 –supplement 2A-D and Figure 1- supplement 3B). Batch correction was required because a similar set of transcriptional clusters (‘Quiescent’, ‘Activated’, ‘Metabolism’, etc) were detected in separate analysis of control, indo and G-CSF samples but not when they were combined without batch correction. Batch correction was only used for dimensionality reduction and clustering, for all differential gene expression analyses (of cluster markers and treatment induced) we used the raw, non-batch corrected counts. We closely followed recommendations by Luecken and Theis (Molecular Systems Biology 15(6) (2019): e8746).

We have updated the manuscript in the following locations to clarify the rationale for dataset integration and batch collections.

1. We have expanded the description in the Results that describes sample processing.

2. We have expanded the batch correction procedure in the Methods.

3. We have added five new analysis Jupyter notebooks (01a – 01e) to GitHub that plot and describe all steps that were required for batch integration.

Was MAST performed comparing 10x version 2 and 10x version 3 captures? This would not be recommended even with combat adjustment, especially since the authors have 10x version 3 data for all captures but not 10x version 2. If the authors are able to replicate their findings with the just the 10x version 3, then the results could be considered higher confidence.

As mentioned above MAST was performed only within 10x vs2 or 10x vs3 capture. All differential gene expression analysis was performed on normalized and log-transformed but non-batch corrected counts.

The authors mention CITE-Seq which implies the use of ADT conjugated antibodies. The reviewer can only find HTO conjugated antibodies to separate rare MPP populations within the same capture (a cost saving method). Moreover, the authors make note that the hashing worked in a prime figure panel (clearly this is a supplementary figure panel). The authors run the risk of conflating cell hashing with informative cell surface antigen detection to the reader. Hence, the term hashing should be used instead of CITE-Seq.

We thank the reviewer for this comment and agree that our original terminology was not fully accurate. We did clarify the terminology throughout the manuscript and replaced CITE-Seq with cell hashing, or HTO labelling. In accordance with the reviewer’s suggestion, we have relegated those results to supplementary material (Figure3 —figure supplement 1C-G).

The analysis of the scATAC seq is extremely limited, and hence the value of the inferred regulators needs to be moderated in the text.

We agree with the reviewer that our scATAC-Seq findings, though intriguing, are rather preliminary. We reorganized and simplified Figure 5 and Figure 5 –supplement 1 in accordance with reviewers #1s suggestion. Furthermore, we have toned down our conclusions around scATAC-Seq results in the abstract, results and conclusions.

Abstract:

‘Chromatin analysis of unperturbed HSCs and LSKs by scATAC-Seq suggested some HSC-specific, cell intrinsic predispositions to niche signals.’

Results:

‘Rather, our analysis implicated cell intrinsic heterogeneity of downstream effectors, such as AP-1 and IRFs that may govern differential transcriptional responses. While cluster enrichment of AP-1 and ISREs was not unique to HSCs we observed a specific occurrence of AP-1 and HSC lineage-specific master factors suggestive of a HSC unique chromatin architecture.’

Discussion

‘Interestingly, we observed heterogeneity of HSC responses to external stimuli which may be determined by the baseline transcriptional and epigenetic state supported by our single cell chromatin studies. Preliminary findings suggested an HSC specific co-occurrence of signaling and lineage-specific transcription factor motif activities that is consistent with previous observations in human hematopoietic progenitors (Trompouki et al., 2011, Choudhuri et al., 2020). Overall, our data indicates that the single cell landscape of in vivo derived, functional HSCs is likely made up of a unique chromatin architecture with fluent transcriptional states, some of which can be rapidly influenced by external signals.’

[Editors' note: further revisions were suggested prior to acceptance, as described below.]

The manuscript has been improved but there are some remaining issues that need to be addressed, as outlined below:The reviewers have agreed on a few sentences that would benefit from further clarification or some edits. Specifically:1) The figure labels seem to be off: Figure 2C does not seem to correspond to the description confirming surface phenotypes with CITE-seq? Description of LSK cells labeled Figure 2C, D, etc in the text (page 6) seems to correspond to the current Figure 3.

We carefully re-examined all figure references throughout the entire manuscript to ensure that they are referencing the correct figure panels.

2) Page 19, Line ~600-there is a discussion of HSC cluster 0 and cluster 1 but these clusters have not been previously introduced. Are they simply based on the ATACseq data, or do they also correspond to some clustering in the RNAseq?

HSC clusters 0 and 1 are solely based on the scATAC-Seq data. We have modified the Results section in the ATAC-seq paragraph to explicitly introduce the clusters.

“We clustered cells based on chromatin accessibility in HSCs resulting in 2 clusters (‘HSC cluster 0’ and ‘HSC cluster 1’, Figure 5B) and LSK cells consisting of MPPs and HSCs resulting in 8 clusters (Figure 5C and Figure 5 —figure supplement 1A-B, Methods).”

3) Discussion:"While we could not detect interferon ligands in our scRNA-Seq data of 723 HSCs or MPPs" – what is meant by "interferon ligands"? Interferons?

The stated wording was changed to the following sentence:

“While we could not detect transcripts for type 1 interferons in our scRNA-Seq data of HSCs or MPPs, it is possible that some of the interferon-response genes were induced indirectly by release of interferons from the niche.”

4) 2-4 orders of magnitude higher than what an animal would ever encounter during… probably one should say "2-4 orders of magnitude higher than what an animal would typically encounter…." "ever" is a strong word.

As suggested, we changed the wording of the above sentence to the following.

"…2-4 orders of magnitude higher than what an animal would typically encounter…”